# Exploring the Role of Novel Biostimulators in Suppressing Oxidative Stress and Reinforcing the Antioxidant Defense Systems in *Cucurbita pepo* Plants Exposed to Cadmium and Lead Toxicity

Mostafa M. Rady [1,*], Mohamed M. M. Salama [1], Sebnem Kuşvuran [2], Alpaslan Kuşvuran [2], Atef F. Ahmed [3], Esmat F. Ali [3,*], Hamada A. Farouk [4], Ashraf Sh. Osman [5], Khaled A. Selim [6] and Amr E. M. Mahmoud [7]

1 Botany Department, Faculty of Agriculture, Fayoum University, Fayoum 63514, Egypt; safycr3@gmail.com
2 Food and Agriculture Vocational School, Cankiri Karatekin University, Çankırı 18100, Turkey; skusvuran@gmail.com (S.K.); akusvuran@gmail.com (A.K.)
3 Department of Biology, College of Science, Taif University, P.O. Box 11099, Taif 21944, Saudi Arabia; atefali@tu.edu.sa
4 Faculty of Agriculture, Department of Plant Pathology, University of Assiut, Assiut 71526, Egypt; hamadafarouk@aun.edu.eg
5 Horticulture Department, Faculty of Agriculture, Fayoum University, Fayoum 63514, Egypt; aso00@fayoum.edu.eg
6 Food Sciences and Technology, Faculty of Agriculture, Fayoum University, Fayoum 63514, Egypt; kas00@fayoum.edu.eg
7 Biochemistry Department, Faculty of Agriculture, Fayoum University, Fayoum 63514, Egypt; aem01@fayoum.edu.eg
* Correspondence: mmr02@fayoum.edu.eg or mrady2050@gmail.com (M.M.R.); a.esmat@tu.edu.sa (E.F.A.)

**Abstract:** The use of bio-stimulants (BSs) has become an important policy in managing many stressed crop plants through the regulation of the balance of phytohormones, osmo-protectors (OPs), antioxidant systems, and gene expression, all of which reflect plant growth and productivity. Garlic + onion extract (GOE) at a concentration of 2.0–3.0% and diluted bee honey solution (BHs) at a concentration of 1.0–1.5% were applied exogenously to squash (*Cucurbita pepo*) plants subjected to cadmium (Cd) + lead (Pb) stress (0.3 mM CdCl$_2$ + 0.3 mM PbCl$_2$). The objective was to determine the effects of these treatments on growth characteristics, organic metabolites/biomolecules, and mineral nutrients. Cd + Pb stress significantly increased electrolyte leakage (EL, 103%) and malondialdehyde (MDA, 90%) because of an increase in hydrogen peroxide (H$_2$O$_2$, 145%) and superoxide (O$_2^{\bullet-}$, 152%) levels, and contents of abscisic acid (ABA, 164%), Cd (674–711%), and Pb (754–805%). Consequently, marked increases in the contents of OPs and non-enzymatic antioxidants (28–133%), activities of antioxidant enzymes (48–80%), and expressions of enzyme genes (60–84%) were observed. The administration of Cd + Pb treatment reduced plant growth and development parameters (25–59%), yield components (61–86%), photosynthetic components (27–67%), leaf proportional water content (26%), indole-3-acetic acid (IAA, 44%), gibberellic acid (GA$_3$, 56%), and cyto-kinin (CKs, 49%) contents. Nonetheless, the administration of GOE, BHs, and GOE + BHs attenuated the adverse impacts of Cd + Pb stress. The best treatment was GOE + BHs which significantly decreased EL (52%) and MDA (49%) because of a reduction of O$_2^{\bullet-}$ (61%), H$_2$O$_2$ (60%), ABA (63%), Cd (89–91%), and Pb (89–91%) levels. This positive outcome was linked to an increase in the OPs' (22–46%) and non-enzymatic antioxidant (27–46%) levels, activities of enzymes (26–44%), and enzyme gene expressions (35–40%), all of which contributed to the promoted relative water content (RWC, 37%), pigment contents (47–194%), hormonal levels (82–132%), growth traits (31–149%), yield components (154–626%), and fruit quality traits (31–92%). From these results, it can be concluded that treatment of GOE + BHs is recommended as a foliar application to reduce the adverse effects of Cd + Pb stress treatment in squash.

**Keywords:** plant extracts; squash; heavy metal stress; oxidants; antioxidants; photosynthesis; osmo-protectors; phytohormones

## 1. Introduction

Environmental pollution caused by heavy metals (HMs) occurs mostly as a result of industrial and agricultural activities. Variable industrial effluents, recurrent applications of sewage sludge, municipal trash, extensive chemical fertilization, and air pollution are the main sources of soil heavy metals. Many of the HMs absorbed by plants in highly contaminated soils are often synergistic, increasing their toxic impacts [1]. Unlike organic substances, the increase of HMs in soil and water because of their non-biodegradability poses significant threats not only to the ecosystem, soil fertility, and water resources but also to animals and humans, which are the last link in the food chain. Depending on the amount of HM in the soil structure, the metabolic pathways in plants change, causing growth limitation, and may lead to plant death [2,3]. HMs are mobile through plasma membranes within the plant and can bind to sulfur and oxygen ($O_2$) and interact directly with proteins and DNA [4,5].

Often, a number of metals are responsible for soil contamination. Cadmium (Cd) and lead (Pb) are among the most toxic HMs in the soil. Cd and Pb are among the most polluting metals in Egyptian soil, including the studied area, resulting from industrial and agricultural activities and car exhausts along agricultural highways [6]. They are listed second and seventh, respectively, in the list of the most dangerous poisonous metals [7]. Maximum permissible levels of soil Cd (1–5 mg kg$^1$) and Pb (50–150 mg kg$^1$) have been recorded for the production of vegetable crops [8]. Because of its tens of thousands of years of persistence in soil, Cd pollution is one of the greatest threats to the agricultural system and animals and human health through the food chain by being absorbed by plant roots [9,10]. Because of the necessary metals and/or cofactors being displaced at the enzyme-active site, metals alter the status of cellular redox. The uptake of Cd has a profound impact on live cells by promoting oxidative stress; hence, depending on the amount and duration of exposure, cell death may result [9,11]. Growing plants exposed to Cd stress exhibit unfavorable changes in several physiological, biochemical, and structural characteristics [12]. Nonspecific indications of Pb poisoning include stunted development, chlorosis, and shorter root lengths. Once within the cell, Pb alters membrane permeability, causes hormonal changes, inhibits enzymes with sulfhydryl groups, reduces water content, and disrupts mineral feeding [13]. The poisonous impacts of HMs have been linked to the production of reactive oxygen species (ROS). Numerous enzymatic and non-enzymatic pathways have been developed in living organisms to quickly get rid of ROS, which cause the disruption of cellular metabolism because of the damage of oxidative stress to important cell molecules. The antioxidant enzymes, such as SOD (superoxide dismutase), CAT (catalase), APX (ascorbate peroxidase), GR (glutathione reductase), along with the non-enzymatic antioxidants, such as AsA (ascorbic acid), $\alpha$ToC ($\alpha$-tocopherol), carotenoids, flavonoids and proline, are included in the antioxidant defense mechanisms and play crucial roles in the detoxification of ROS and in maintaining the metal stress tolerance in plants, including wheat (cvs. Sakha 93 and Pradip), *Arabidopsis thaliana* (wild type (Col. 0) and mutant *AtrbohC*, *AtrbohD* and *AtrbohF*), cucumber (var. Wisconsin), and menthol mint (cv. Kosi and Kushal) [14,15].

Depending on its nutrient and antioxidant content, a bio-stimulant is any microbe or chemical used to increase the nutritional value, resistance to abiotic stresses, including heavy metals (HMs), and/or the qualitative features of plants, including maize (cv. Hybrid 306), pepper (cv. Top Star), wheat (cv. Misr 2), and chili pepper [10,16–18]. Bio-stimulants (BSs) increase the efficiency of nutrient use in plants, provide tolerance against abiotic stress conditions, improve product quality, and contribute to the effective use of nutrients found in limited amounts in the soil and root zone by plants. The use of natural plant BSs obtained

by extraction from organic raw materials containing bioactive compounds increases the growth, flowering, fertilization, fruit set, productivity, and nutrient utilization efficiency in plants, including maize (cv. Hybrid 306), sweet sorghum (var. local), and *Atriplex nummularia*, contributing to an increase of tolerance against abiotic stress factors [10,19,20]. The presence of various phytohormones, vitamins, antioxidants, organic and inorganic nutrients in the BSs can directly affect plant growth and production by increasing the stress tolerance in plants, including pepper (cv. Top Star) and chili pepper [21,22]. Many techniques have been recently developed to assess the antioxidant potential of BSs on important crops [23,24]. The performance of plants under stress circumstances, including HM stress, may be improved by the bioactive substances found in a variety of natural BSs [10].

The use of BSs to alleviate stresses, including from HMs, has been widely studied and applied. Among these are microbial BSs, but there are some limitations that explain the difficulty of using these. Before using microbial BSs, the relationship between the host plant and the microorganism must be evaluated. It cannot be said with certainty that any microbe is universally suitable for all ecosystems or for any host plant. For the development and use of microbial BSs, many factors must be taken into account, including crop, soil, and microbial strain characteristics, as well as the type of application. The selection of a strain to produce microbial BSs needs plant and soil characteristics to be considered to maximize efficacy under specific conditions [25]. However, there are some plant BSs that can be easily prepared and used regardless of the above limitations, including garlic extract (GE), onion extract (OE), garlic + onion extracts (GOE), and diluted bee honey solution (BHs).

One of the BSs with stimulating and antifungal qualities is GOE. Because of the high concentration of biological compounds—more than 200—in these extracts, including vitamins and antioxidants, it is recognized as very nutritious. Powerful antioxidants, including organosulfur compounds such as allicin, diallyl disulfide, and diallyl trisulfide, are antifungal, antibacterial, and antiviral activities of GE and OE [26]. In the reports [27–29], GE and OE have been applied as foliar spraying approaches for Anna apple dormant buds and rats. These reports indicated that OE has the potential to hasten dormancy release in buds through enhancing carbohydrate metabolism, increasing reducing sugars and hydrolytic enzyme ($\alpha$-amylase and invertase) activities. Additionally, GE has the potential to hasten dormancy release in buds through improving bud water content, total carbohydrates, reducing and total sugars, anthocyanins, total free amino acids, free proline, and total indoles. These positive results due to OE or GE treatment hasten the date of floral bud break and increase the percentage of bud break, fruit set, total number of fruits, and fruit yield per tree.

Mono-, di-, and polysaccharides, as well as nutrients, vitamins, fatty acids, proteins, enzymes, organic acids, and amino acids, are all present in BHs. Glucose in BHs is converted by the enzyme glucose oxidase into gluconic acid and hydrogen peroxide ($H_2O_2$), both of which have proven antibacterial properties [30]. In BHs, the CAT enzyme reacts with $H_2O_2$ to release nascent oxygen. BHs has antioxidant and free radical scavenging properties [30,31]. In the reports [10,30–32], BHs (as one of the powerful BSs) has been applied as a foliar spraying strategy for *Atriplex nummularia* seedlings grown under saline-calcareous stress, faba bean plants (cv. Giza 40) grown under drought stress, onion plants (cvs. Giza 20 and Giza$^{Red}$) grown under salt stress, and common bean plants (cv. Bronco) grown under drought stress, respectively. These reports signaled that BHs has the potential to increase stress tolerance in plants. This increased stress tolerance in plants is associated with a marked improvement in growth and productivity traits, yield quality traits, attributes related to photosynthetic efficiency, leaf integrity, nutritional and hormonal balances, components related to osmoregulation, enzymatic and non-enzymatic antioxidant systems, and enzyme gene expressions, all accompanied by noticeable suppression of oxidative stress markers ($O_2^{\bullet-}$ and $H_2O_2$), and toxic ions [10,30–32].

Squash (*Cucurbita pepo* L.) is a fundamental vegetable and commercial crop globally. One-third of all squash output worldwide is from Egypt, Italy, and Turkey [33]. Most

of the research has used short-term trials to examine how BSs affect plant growth and production. The use of BHs combined with garlic + onion extract (GOE) as multiple BSs to counteract the adverse impacts of Cd + Pb stress on squash plants has not yet been studied. In the present study, the development and production of squash plants cultivated under Cd + Pb stress with GOE, BHs, or GOE + BHs applied as foliar feeding were examined. The major objectives of this research were to reveal the impacts of supplementing Cd + Pb-exposed squash plants with GOE, BHs, or GOE + BHs on ROS levels, activities of different antioxidant (non-enzymatic and enzymatic), and gene expression, photosynthetic efficiency and osmotic adaptation, hormonal balance, and squash growth and productivity, and fruit quality. Based on the previous studies on GE, OE, and BHs applied individually and their efficiency to attenuate stress impacts in plants [27–32], this study hypothesized that the application of BHs combined with GOE as a foliar spraying strategy will increase the tolerance to Cd + Pb stress in squash plants causing an increase in plant growth, yield quality due to improvement in photosynthetic efficiency, antioxidant (non-enzymatic and enzymatic) activities, enzyme gene expression, osmotic adaptation, and hormonal balance.

## 2. Materials and Methods

### 2.1. Study Site, Plant Material, and Water Analysis

For the duration of March−May 2022, appropriate plastic containers (42 cm diameter, 40 cm depth, 6.0 L) were used to conduct an experiment in a wire greenhouse located at Demo farm (29°17′ N; 30°53′ E), Fayoum, Egypt. The study region is categorized as an arid environment according to the aridity index suggested by Ponce et al. [34]. Table S1 reveals the climatic conditions of the study region during the two growing seasons.

Hybrid Hi Tech® squash seeds (Misr Hi-Tech International Seed Company, Cairo, Egypt) were purchased for these trials. One hundred and sixty plastic containers were filled with the growth medium described in Rady and Rehman [35] with modification. The medium was supplemented with fully decomposed farmyard manure (FYM) and compost. The basic components of the medium were peat moss (PM), vermiculite (Vct), crushed maize grains (CMG), and humic acid (HA) [35]. The modified medium consisted of 15.0% PM, 10.0% Vct, 6.0% CMG, 0.5% HA, 15.0% FYM, and 3.5% compost (by weight). The nutritional content of this modified medium was 13.24, 0.32, 8.22, 3.12, 3.04, 1.66, 2.44, 1.62, and 0.69 g kg$^{-1}$ of N, P, K, Ca, Mg, S, Fe, Mn, and Zn, respectively. The following formulation of fertilizer was mixed well with the modified medium: 0.415 g $NH_4NO_3$ L$^{-1}$; 0.50 g $CaH_6O_9P_2$ L$^{-1}$; 0.333 g $K_2SO_4$ L$^{-1}$; 0.833 g $MgSO_4$ L$^{-1}$; 0.333 mg $Fe^{2+}$ L$^{-1}$; 0.333 g Zn L$^{-1}$; and 0.333 g Mn L$^{-1}$. To adjust the pH, 1.25 g of $CaCO_3$ was added L$^{-1}$ of the modified medium. Additionally, per L of the modified medium, 0.125 g fungicide Moncut SC (wettable powder, Central Glass Co., Ltd., Tokyo, Japan) was added to prevent the growth of pathogens [22,36]. In each container, one seed was planted (3 cm deep in the middle of the modified medium) on 23 March and harvesting ended on 20 May. The same fertilizer formulation was also added to the modified medium 25 days after planting (DaP). The plants were irrigated regularly (at modified medium field capacity) every three days. Table S2 reveals the analysis of the water used in this study.

### 2.2. Experimental Layout

A complete randomized block design was applied to arrange the study treatments with five replications (each replicate contained 4 pots). The experiment was repeated 3 times in the same experimental region at the same time. For the non-contaminated medium (nCM) and contaminated medium (CM), eight treatments were designed, four treatments for each (nCM or CM). Description of the study treatments and foliar spraying times are presented in Table 1. At the same concentrations, garlic + onion extract (GOE) exceeded garlic or onion extract on a preliminary trial basis. In addition, the concentrations of GOE and bee-honey solution (BHs), as well as the number of sprays of both GOE and BHs were selected on a preliminary trial basis. Using all spraying solutions (and distilled water for the control plants), the seedlings/plants were sprayed early in the morning to run-off. An average of 40 mL of spraying solution was applied to each plant at each spraying

time. Appropriate surfactant Tween-20 (0.1%, $v/v$) was mixed with the spray solution to maximize optimal solution penetration into the leaves of squash plants. Different cultural practices, including disease and pest control management, were followed for commercial squash production.

**Table 1.** Description of the study treatments and foliar spray times.

| Treatments | | Description |
|---|---|---|
| **Cd + Pb** | **Bio-Stimulator** | |
| 0 mM | Control | No stress + Three foliar sprays of distilled water 15, 22, and 29 days after transplanting (DaT). |
| | GOE | No stress + Three foliar sprays of 3.0% GOE (3-FS.GOE) 15, 22, and 29 DaT. |
| | BHs | No stress + Three foliar sprays of 1.5% BHs (3-FS.BHs) 15, 22, and 29 DaT. |
| | GOE + BHs | No stress + Two foliar sprays of 2.0% GOE (2-FS.GOE) 12 and 24 DaT + Two foliar sprays of 1.0% BHs (2-FS.BHs) 18 and 30 DaT. |
| 0.3 mM + 0.3 mM | Control | Irrigating squash plants with water containing 0.3 mM Cd + 0.3 mM Pb throughout the experiments. |
| | GOE | Irrigating squash plants with water containing 0.3 mM Cd + 0.3 mM Pb throughout the experiments + 3-FS.GOE 15, 22, and 29 DaT. |
| | BHs | Irrigating squash plants with water containing 0.3 mM Cd + 0.3 mM Pb throughout the experiments + 3-FS.BHs 15, 22, and 29 DaT. |
| | GOE + BHs | Irrigating squash plants with water containing 0.3 mM Cd + 0.3 mM Pb throughout the experiments + 2-FS.GOE 12 and 24 DaT + 2-FS.BHs 18 and 30 DaT. |

GOE; a mixture of garlic and onion extracts at equal volumes and concentrations. BHs; bee-honey solution. Cd; cadmium, and Pb; lead. Cd + Pb; both Cd and Pb were added at equal concentrations (0.3 mM each) using $CdSO_4$ and $PbSO_4$.

### 2.3. Preparation of Garlic Extract (GE), Onion Extract (OE) and Bee-Honey Solution (BHs)

Following the detailed procedures described in Rady and Seif El-Yazal [27,28], Egyptian white garlic cloves and onion heads (Giza-20) were used to prepare GE and OE. After peeling, the cloves or onion heads were ground in a large mortar and pestle with ethanol (95%) to extract the active ingredients. The ethanol mixture was filtered and then evaporated under vacuum (30 °C) using a rotary evaporator (Buchi model 011) to remove the alcohol. The resulting paste is expressed as GE or OE and used immediately (or kept cold at −20 °C until use). GE was diluted with distilled water (dW) to obtain the desired concentrations (3%; 30 g paste $1^{-1}$ dW, and 2%; 20 g paste $1^{-1}$ dW) immediately before use. Like the GE concentrations, OE concentrations 3 and 2% were prepared. The same volumes of both extracts were mixed to obtain garlic + onion extract (GOE) at 3 and 2% concentrations, because the best response compared to GE or OE was generated on the basis of the initial experiment. Table S3 reveals the analysis of the bioactive components of GOE. To prepare the BHs, 10 g or 15 g pure honey was well dissolved in 1 L dW to prepare the 1.0% or 1.5% BHs. Table S4 reveals the analysis of the bioactive components of honey.

### 2.4. Sampling

For different assessments of plant morphology, physiology, and biochemistry, five plants were chosen at random from each treatment 10 days after the last spraying (40 days after planting; start time of first fruit harvest). For different components of the antioxidant defensive systems and phytohormone determinations, another five plants were chosen at

random from each treatment. The remaining 10 pots were allocated to the yield components. Fruit harvesting started 40 days after planting until the trial termination.

## 2.5. Evaluation of the Efficiency of Photosynthesis

The content of chlorophylls and carotenoids was determined in the leaves of squash [37]. The homogenization solution used was 80% acetone and after obtaining the homogenates, they were centrifuged for 10-min at $10,000 \times g$. Wavelengths 663, 645, and 470 nm were identified for the absorbance readings. Using the chlorophyll-meter (Minolta, Osaka, Japan), the SPAD index was assessed. The efficiency of photosynthesis; Fv/Fm and performance index (PI) were assessed [38,39]. The procedures of Jagendorf [40] and Avron [41] were performed to assess the photochemical activity in squash fresh leaves.

## 2.6. Assessment of Leaf Integrity, Levels of Cadmium (Cd), Lead (Pb), and Oxidative Stress Markers

Applying the procedures and equations depicted in Osman and Rady [42], Rady [43], and Rady and Rehman [35], leaf relative water content, membrane stability index and leakage, respectively, were assessed. Following the procedures depicted in Chapman and Pratt [44], root and leaf contents of Cd and Pb (mg kg$^{-1}$ DW) were analyzed in roots and leaves using an Atomic Absorption Spectrophotometer (Perkin-Elmer, Hong Kong, Chian, Model 3300). Then, 0.1 g dried root and leaf samples were digested with an acid mixture [1 80% perchloric acid: 2 concentrated $H_2SO_4$ ($v/v$)] for 12 h. After dilution and addition of distilled water reaching a volume of 100 mL, the contents of $Cd^{2+}$ and $Pb^{2+}$ were measured. Applying the procedures of Kubiś [45], Velikova et al. [46], and Madhava Rao and Sresty [47], the levels of $H_2O_2$, $O_2^{\bullet-}$, and malondialdehyde expressing lipid peroxidation, respectively, were assessed.

## 2.7. Assessment of Osmo-Protectors and Antioxidant Contents

The contents of glycine betaine (GB; µM g$^{-1}$ DW), soluble sugars (mg g$^{-1}$ DW), proline (µM g$^{-1}$ DW), ascorbate (AsA; µM g$^{-1}$ DW), glutathione (GSH; µM g$^{-1}$ DW), and $\alpha$-tocopherol ($\alpha$ToC; µM g$^{-1}$ DW) were assessed following the procedures of Grieve and Grattan [48], Irigoyen et al. [49], Bates et al. [50], Huang et al. [51], Paradiso et al. [52], and Baker et al. [53], respectively. GB content was measured colorimetrically (at 365 nm) using a cold-KI-$I_2$ as a reagent. Soluble sugar content was measured colorimetrically (at 625 nm) using anthrone as a reagent freshly prepared in 72% sulfuric acid. Proline content was measured colorimetrically (at 520 nm) using toluene solution for extraction. AsA content was measured colorimetrically (at 530 nm) after homogenization using 1 mM EDTA in HPO$_3$ solution (5%). GSH content was measured colorimetrically (at 412 nm) after adding GSH reductase. $\alpha$ToC content was measured colorimetrically (at 520 nm) after homogenizing the samples in ethanol and petroleum (1.6:2, $v/v$) and then mixing the centrifuged supernatant with 2-dipyridyl in ethanol.

## 2.8. Assaying Enzyme Activities and Enzyme Gene Expressions

Enzyme extracts (EEs) were obtained after homogenizing 0.2 g of freeze-dried leaf samples, each in 2 mL K-P buffer (100 mM, pH 7.0). One hundred mM EDTA solution was added to the extraction buffer (EB). The EB also received 2 mM AsA only in the APX activity assay. After filtering, the filtrates were centrifuged for $\frac{1}{4}$ h at $12,000 \times g$. If the EEs were not to be used immediately, they were stored at 25°C until use. All the previous steps were conducted under refrigeration (4 °C).

Expressed in Unit g$^{-1}$ protein, SOD activity was assayed, as were CAT, APX, and GR, all expressed in µmol $H_2O_2$ min$^{-1}$ g$^{-1}$ protein and assayed following the procedures of Beauchamp and Fridovich [54], Havir and McHale [55], Nakano and Asada [56], and Foster and Hess [57], respectively. Soluble protein content (mg g$^{-1}$ DW) was assessed following the procedures of Bradford [58].

Squash leaf samples were used for total RNA isolation by using a RNeasy Mini Kit (Qiagen GmbH, Hilden, Germany). Then, the posterior cDNA was synthesized with the use of a RevertAid H Minus First Strand cDNA Synthesis Kit (Fermentas GmbH, Leon-Rot, Germany). Table S5 reveals the primer sequences for quantitative and semi-quantitative RT-PCR of the stress-related genes in squash. Manufacturer's directives of iQ SYBR Green Supermix (Bio-Rad, Hercules, CA, USA) were advised to analyze the quantitative PCR (qRT–PCR) on the iCycler Thermal Cycler (Bio-Rad, USA). As a reference, two actin genes were used for qPCR data normalization. LinRegPCR Software, version 7.4 was used to compute the reaction efficiency [59]. Then, the Pfaffl [60] equation was used to derive signal values from the threshold cycles, subtracting the background mean.

### 2.9. Assessment of Hormonal Content

The GC-MS procedures were applied to evaluate indole-3-acetic acid (IAA), gibberellic acid ($GA_3$), and cyto-kinin (CK) profiling [61,62]. Fresh leafy samples, 100 mg each, were extracted each in 2 mL (ice-cold) of 80% $CH_3OH$: 19.9% $H_2O$: 0.1% 6 N HCl ($v/v/v$). At 4 °C, the resulting extracts were centrifuged for 5-min at $25,000\times g$. After the supernatants were collected, they were concentrated to 50 μL each under a stream of N and then stored under −80 °C until use. For IAA, the supernatant (50 μL) was derivatized with MCF (40 μL), and under stream of N it was concentrated to 20 μL. The organic phase was then dried by adding 0.5 mg of $Na_2SO_4$. For both $GA_3$ and CKs, the supernatants (50 μL each) were dried and derivatized with N-Methyl-N-(trimethylsilyl)trifluoroacetamide (MSTFA) (100 μL) under 85 °C for $\frac{3}{4}$ h. Then, hormonal analysis was performed using GC–MS (Perkin Elmer, Waltham, MA, USA). The IAA, $GA_3$, CKs were identified by comparing hormonal retention time, linear retention indices (LRIs), and selected ions with authentic standards. Extraction and determination of ABA levels were performed using HPLC [63].

### 2.10. Assessment of Growth, Yield, and Fruit Quality Characteristics

Forty days after cultivation, 9 plants were randomly chosen from each treatment to evaluate growth characteristics. Shoot length was assessed using a large ruler. After counting the leaves on each of the 9 plants, their areas were taken utilizing a digital planimeter (Planix 7). The shoot of each plant was weighed to record fresh weights; then dry weights were taken after drying the shoots at 70 °C until reaching stable weights. At the same time (40 days after cultivation), fruits from each of the remaining squash plants were picked. Then, the fruit picking was continued until the fruiting was ended to record the average weight (g) per fruit, fruits number per plant, and fruits weight (kg) per plant (the yield).

Fresh squash fruits were exposed to quality assessments. Expressed in mg per 100 g fresh fruits, vitamin C content was evaluated, as were total soluble solids (°Brix), titratable acidity (%), soluble sugars (% DW), and metal (Cd and Pb; mg per kg dried fruits) contents following the methods of Law et al. [64], A.O.A.C. [65], Irigoyen et al. [49], and Chapman and Pratt [44], respectively.

### 2.11. Data Analysis

The findings of the three trials were subjected to a combined analysis using mixed models, and tested for error variance homogeneity [66]. Then, the two-way ANOVA was applied to analyze all data statistically through the GLM procedure of Gen STAT (version 11) (VSN International Ltd., Oxford, UK). The Turkey's test was applied to test the differences between means [67] at the 0.01 ($p \leq 0.01$) probability level. The data presented in Figures are means ± SE (standard error). Pearson's correlation coefficients, heatmap, and PCA biplot graphs were carried out with the utilization of the R software (version 4.1.3, https://CRAN.R-project.org, accessed on 15 January 2023).

## 3. Results

### 3.1. Photosynthetic Activity of Plants

As shown in Figure 1, under normal conditions, foliar spraying of bio-stimulators (BSs; GOE, BHs, or GOE + BHs) significantly increased SPAD value (Soil Plant Analysis Development), total chlorophyll content (TChC), total carotenoids content (TCrC), PSII efficiency (Fv/Fm), performance index (PI), and photo-chemical activity (PhA) of squash plants. The best treatment was GOE + BHs, which increased these traits by 23, 28, 24,10, 31, and 22%, respectively, relative to the control (foliar spraying with distilled water). Stress induced from 0.3 mM cadmium + 0.3 mM lead (Cd + Pb) markedly decreased SPAD value, TChC, TCrC, Fv/Fm, PI, and PhA by 51, 67, 29, 27, 30, and 41%, respectively, relative to the control. However, the application of different BSs (GOE, BHs, or GOE + BHs) restored leaf photosynthetic efficiency-related traits of squash plants grown under Cd + Pb stress. The best treatment was GOE + BHs, which increased SPAD value, TChC, TCrC, Fv/Fm, PI, and PhA by 107, 194, 47, 36, 46, and 70%, respectively, relative to the stressed control. The efficiency of the GOE + BHs treatment in promoting the leaf photosynthetic efficiency-related traits under Cd + Pb stress was greater than that under normal conditions.

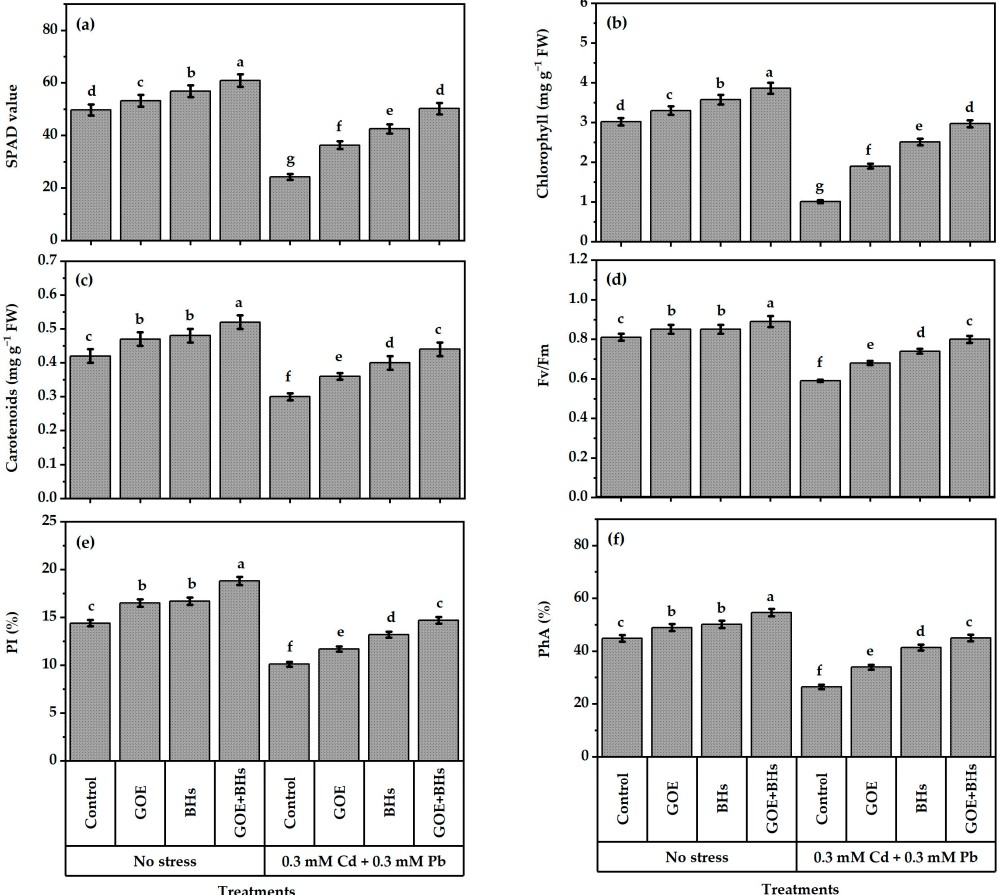

**Figure 1.** Responses of photosynthetic efficiency traits [(**a**) SPAD value; Soil Plant Analysis Development, (**b**) total chlorophyll content, (**c**), total carotenoids content, (**d**), PSII efficiency; Fv/Fm, (**e**) performance index; PI, and (**f**) photo-chemical activity; PhA] of *Cucurbita pepo* plants to foliar spraying with biostimulators (BSs) under normal or heavy metal [cadmium (Cd) + lead (Pb)] stress conditions. In each column, each two means followed by different letters indicate a significant difference according to the LSD test ($p \leq 0.05$). GOE; garlic + onion extract, BHs; bee-honey solution.

### 3.2. Leaf Tissue Stability

As displayed in Figure 2, under normal conditions, foliar spraying of different BSs (GOE, BHs, or GOE + BHs) markedly increased the leaf relative water content (RWC) and

membrane stability index (MSI) of squash plants. The best treatment was GOE + BHs, which increased RWC by 9% and MSI by 17% compared to the control. Combined (Cd + Pb) stress markedly decreased RWC and MSI by 26 and 24%, respectively, compared to the control. However, the application of different BSs (GOE, BHs, or GOE + BHs) restored leaf RWC and MSI of squash plants grown under Cd + Pb stress. The best treatment was GOE + BHs, which increased RWC and MSI by 37 and 33%, respectively, relative to the stressed control. The efficiency of the GOE + BHs treatment in restoring leaf integrity under Cd + Pb stress was greater than that under normal conditions.

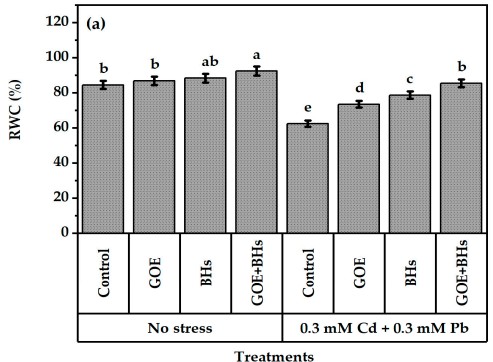 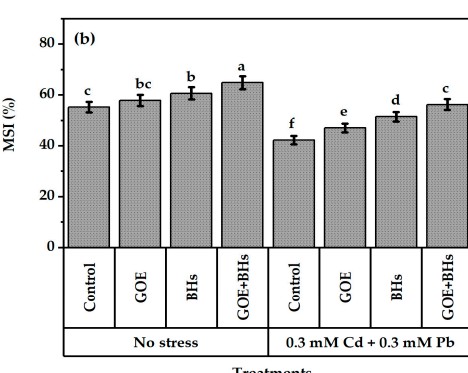

**Figure 2.** Responses of leaf cell integrity ((**a**) relative water content; RWC and (**b**) membrane stability index; MSI) of *Cucurbita pepo* plants to foliar spraying with biostimulators (BSs) under normal or heavy metal (cadmium (Cd) + lead (Pb)) stress conditions. In each column, each two means followed by different letters indicate a significant difference according to the LSD test ($p \leq 0.05$). GOE; garlic + onion extract, BHs; bee-honey solution, RWC; relative water content, and MSI; membrane stability index.

### 3.3. Content of Cadmium (Cd) and Lead (Pb) in Leaves and Roots

As depicted in Figure 3, under normal conditions, foliar spraying of GOE, BHs, or GOE + BHs noticeably decreased the root and leaf contents of Cd and Pb in squash plants. The best treatment was GOE + BHs, which decreased root Cd content (R-Cd) by 43%, leaf Cd content (L-Cd) by 28%, root Pb content (R-Pb) by 39%, and leaf Pb content (L-Pb) by 27% compared to the control. Cd + Pb stress markedly increased R-Cd, L-Cd, R-Pb, and L-Pb by 674, 754, 711, and 805%, respectively, compared to the control. However, the application of GOE, BHs, or GOE + BHs noticeably minimized these HMs in the squash roots and leaves under Cd + Pb stress. The best treatment was GOE + BHs, which decreased R-Cd, L-Cd, R-Pb, and L-Pb by 89, 89, 91, and 91%, respectively, compared to the stressed control. The promoting effect of BSs, especially GOE + BHs, was more pronounced under stress than under no-stress conditions.

### 3.4. Membrane Damage Linked to Oxidative Stress and Indicators of Oxidative Stress

Figure 4 declares that, under normal conditions, applying GOE, BHs, or GOE + BHs as foliar spray markedly decreased the levels of oxidative stress markers ($H_2O_2$ and $O_2^{\bullet-}$) and their damage in terms of electrolyte leakage (EL) and malondialdehyde (MDA) content in squash leaves. The best treatment was GOE + BHs, which decreased $H_2O_2$, $O_2^{\bullet-}$, EL, and MDA levels by 35, 26, 29, and 27%, respectively, compared to the control. Cd + Pb stress markedly increased oxidative stress markers and their damage; $H_2O_2$, $O_2^{\bullet-}$, EL, and MDA levels by 145, 152, 103, and 90%, respectively, compared to the control. However, foliar spraying of GOE, BHs, or GOE + BHs noticeably minimized these oxidative stress markers and their damage in the squash leaves under Cd + Pb stress. The best treatment was GOE + BHs, which decreased $H_2O_2$, $O_2^{\bullet-}$, EL, and MDA levels by 60, 61, 52, and 49%, respectively, compared to the stressed control. The efficiency of the combined (GOE + BHs) treatment in minimizing the oxidative stress markers and their damage under Cd + Pb stress was greater than that under normal conditions.

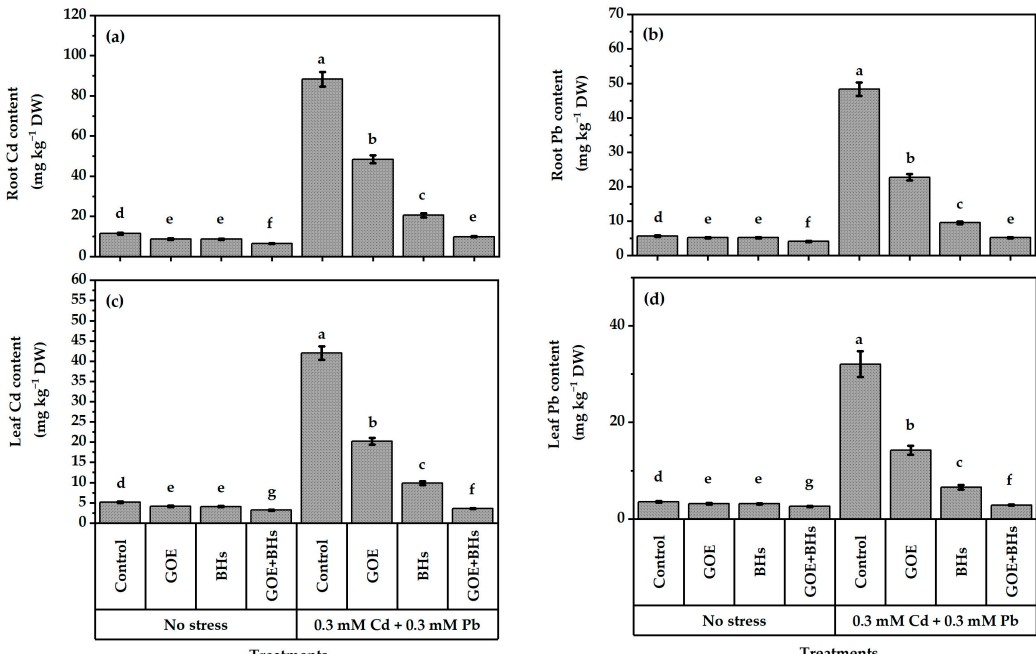

**Figure 3.** Responses of (**a**) root Cd content, (**b**) root Pb content, (**c**) leaf Cd content, and (**d**) leaf Pb content of *Cucurbita pepo* plants to foliar spraying with biostimulators (BSs) under normal or heavy metal [cadmium (Cd) + lead (Pb)] stress conditions. In each column, each two means followed by different letters indicate a significant difference according to the LSD test ($p \leq 0.05$). GOE; garlic + onion extract and BHs; bee-honey solution.

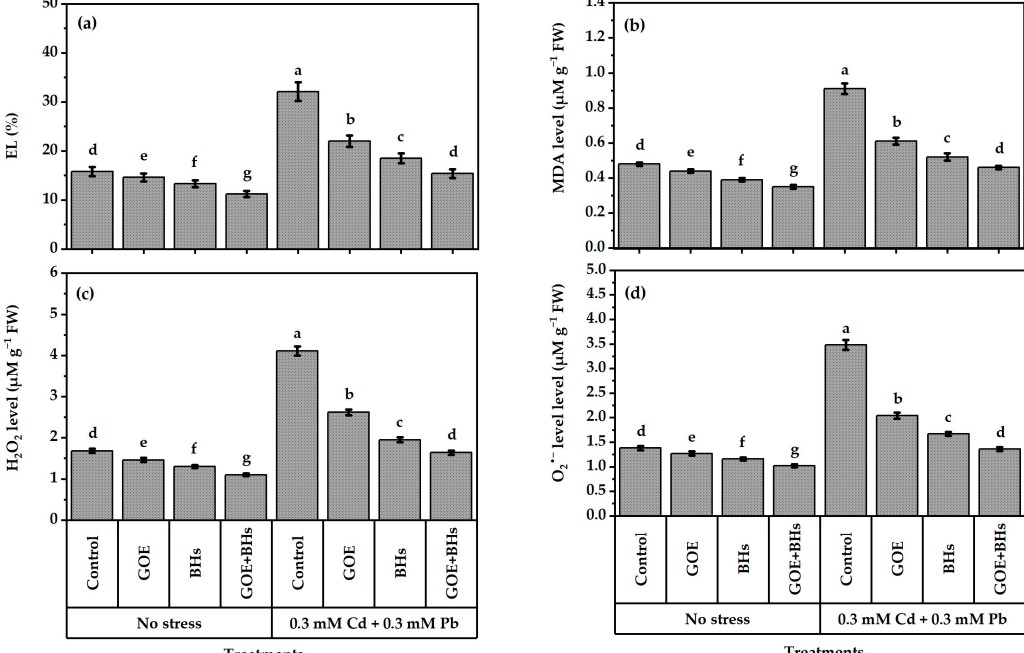

**Figure 4.** Responses of (**a**) leaf electrolyte leakage (EL), (**b**) lipid peroxidation in terms of malondialdehyde (MDA) content, (**c**) hydrogen peroxide ($H_2O_2$) level, and (**d**) superoxide ($O_2^{\bullet-}$) level of *Cucurbita pepo* plants to foliar spraying with biostimulators (BSs) under normal or heavy metal [cadmium (Cd) + lead (Pb)] stress conditions. In each column, each two means followed by different letters indicate a significant difference according to the LSD test ($p \leq 0.05$). GOE; garlic + onion extract, and BHs; bee-honey solution.

### 3.5. Osmo-Protectors (OPs) and Non-Enzymatic Antioxidants

Figure 5 shows that, under normal conditions, applying GOE, BHs, or GOE + BHs as foliar spray markedly increased the contents of osmo-protectors (OPs) and non-enzymatic antioxidants (glycine betaine; GB, soluble sugars, proline, ascorbate; AsA, glutathione; GSH, and α-tocopherol; αToC) in squash leaves. The best treatment was GOE + BHs, which increased GB, soluble sugars, proline, AsA, GSH, and αToC contents by 21, 23, 58, 37, 61, and 27%, respectively, compared to the control. Cd + Pb stress also increased the contents of OPs and non-enzymatic antioxidants by 23, 37, 133, 54, 64, and 28%, respectively, compared to the control. Additionally, foliar spraying of GOE, BHs, or GOE + BHs noticeably maximized the contents of OPs and non-enzymatic antioxidants in the squash leaves under Cd + Pb stress. The best treatment was GOE + BHs, which increased GB, soluble sugars, proline, AsA, GSH, and αToC contents by 22, 26, 46, 33, 37, and 27%, respectively, compared to the stressed control.

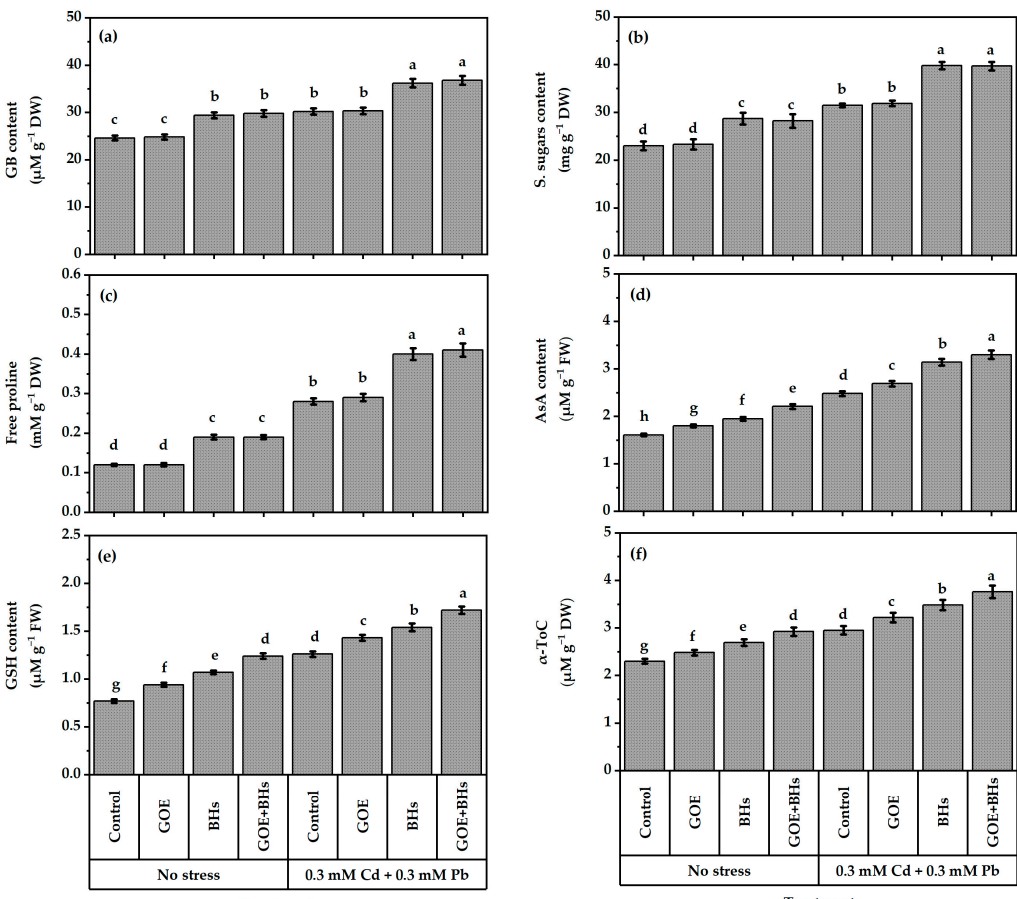

**Figure 5.** Responses of osmo-protectant and antioxidant compounds contents ((**a**) glycine betaine; GB, (**b**) soluble sugars, (**c**) free proline, (**d**) ascorbate; AsA, (**e**) glutathione; GSH, and (**f**) α-Tocopherol; αToC) of *Cucurbita pepo* plants to foliar spraying with biostimulators (BSs) under normal or heavy metal (cadmium (Cd) + lead (Pb)) stress conditions. In each column, each two means followed by different letters indicate a significant difference according to the LSD test ($p \leq 0.05$). GOE; garlic + onion extract and BHs; bee-honey solution.

### 3.6. Antioxidant Defense System Components

As shown in Figures 6 and 7, under normal conditions, applying GOE, BHs, or GOE + BHs as foliar spray markedly increased the activities of antioxidant enzymes (SOD, CAT, APX, and GR) and enzyme gene expressions (*SOD, CAT, APX, GR,* and *PrxQ*) in squash plants. The best treatment was GOE + BHs, which increased antioxidant enzyme activities and enzyme gene expressions by 36, 59, 46, 56, 52, 58, 62, 50, and 74%, respectively,

and soluble protein content was also increased by 32% compared to the control. Cd + Pb stress also increased SOD, CAT, APX, and GR activities by 37, 80, 48, and 78%, respectively, as well as gene (*SOD, CAT, APX, GR,* and *PrxQ*) expressions by 73, 60, 81, 69, and 84%, respectively, while soluble protein content was decreased by 35% compared to the control. Additionally, foliar spraying of GOE, BHs, or GOE + BHs noticeably maximized the activities of antioxidant enzymes and enzyme gene expressions in the squash leaves under Cd + Pb stress. The best treatment was GOE + BHs, which increased SOD, CAT, APX, and GR activities and soluble protein content by 26, 39, 44, 38, and 53%, respectively, as well as gene (*SOD, CAT, APX, GR,* and *PrxQ*) expressions by 38, 39, 35, 38, and 40%, respectively, compared to the stressed control.

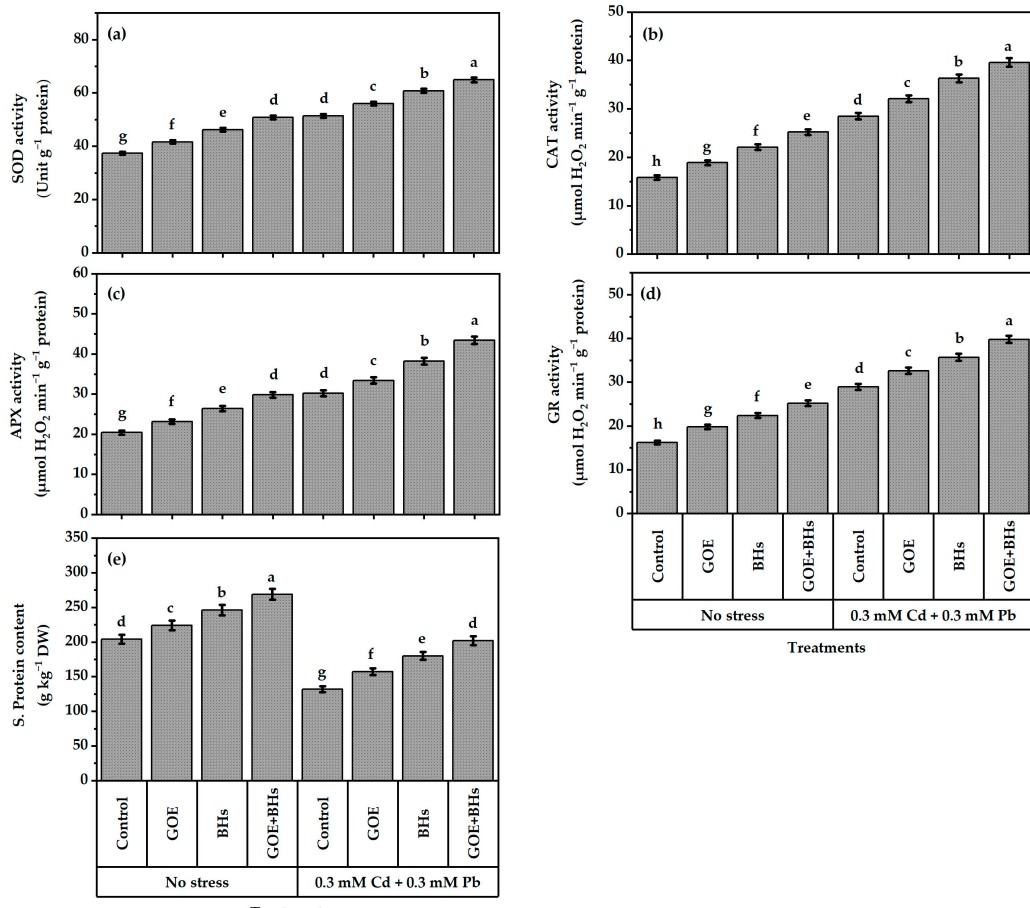

**Figure 6.** Responses of enzymatic antioxidant activities ((**a**) superoxide dismutase; SOD, (**b**) catalase; CAT, (**c**) ascorbate peroxidase; APX, and (**d**) glutathione reductase; GR) and (**e**) soluble protein content of *Cucurbita pepo* plants to foliar spraying with biostimulators (BSs) under normal or heavy metal (cadmium (Cd) + lead (Pb)) stress conditions. In each column, each two means followed by different letters indicate a significant difference according to the LSD test ($p \leq 0.05$). GOE; garlic + onion extract. BHs; bee-honey solution.

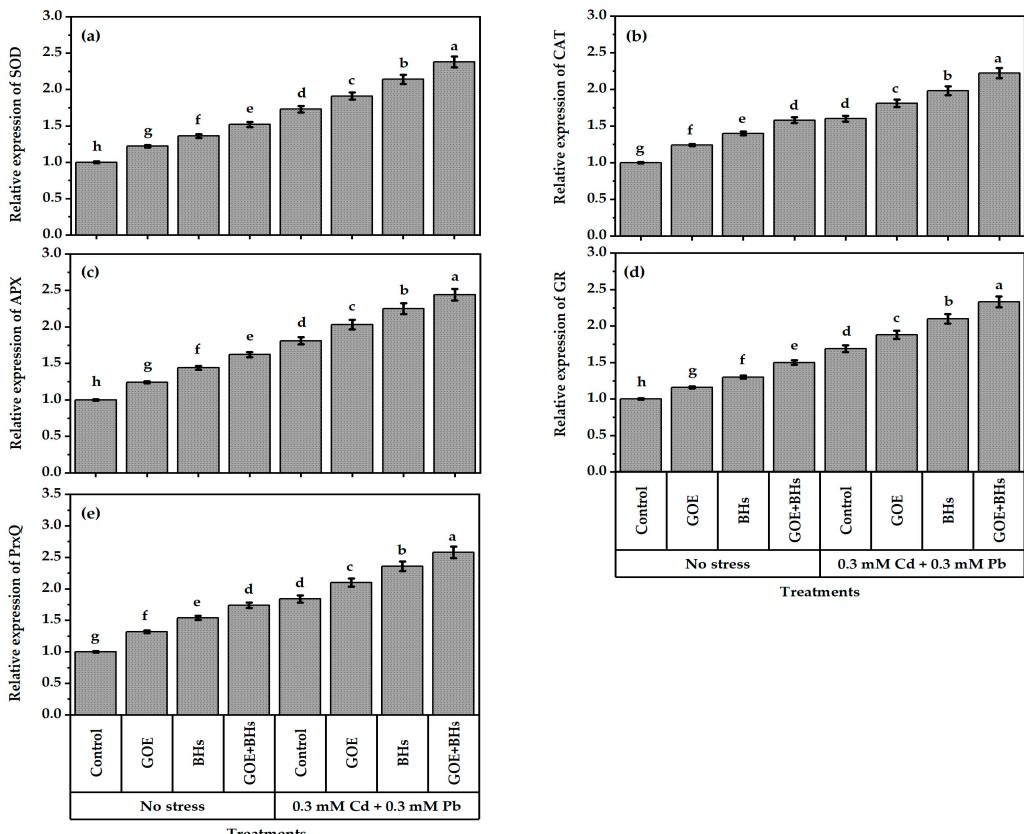

**Figure 7.** Responses of transcript levels of antioxidant enzyme encoding genes ((**a**) *SOD*, (**b**) *CAT*, (**c**) *APX*, (**d**) *GR*, and (**e**) *PrxQ*; peroxiredoxin) of *Cucurbita pepo* plants to foliar spraying with bios-timulators (BSs) under normal or heavy metal (cadmium (Cd) + lead (Pb)) stress conditions. In each column, each two means followed by different letters indicate a significant difference according to the LSD test ($p \leq 0.05$). GOE; garlic + onion extract. BHs; bee-honey solution.

### 3.7. Content of Phytohormones in Plant Leaves

As shown in Figure 8, under normal conditions, foliar spraying of bio-stimulators (BSs; GOE, BHs or GOE + BHs) significantly increased the contents of indole-3-acetic acid (IAA), gibberellic acid (GA$_3$), and cyto-kinins (CKs), while abscisic acid (ABA) content was decreased in squash leaves. The best treatment was GOE + BHs, which increased IAA, GA$_3$, and CKs by 33, 40, and 35%, respectively, while ABA decreased by 48% relative to the control (foliar spraying with distilled water). Combined (Cd + Pb) stress markedly decreased the contents of IAA, GA$_3$, and CKs by 44, 56, and 49%, respectively, while ABA increased by 164% relative to the control. However, the application of different BSs (GOE, BHs, or GOE + BHs) restored leaf phytohormone homeostasis in squash plants grown under Cd + Pb stress. The best treatment was GOE + BHs, which increased IAA, GA$_3$, and CKs by 82, 132, and 94%, respectively, while ABA was decreased by 63% relative to the stressed control. The efficiency of the GOE + BHs treatment in balancing the leaf phytohormones under Cd + Pb stress was greater than that under normal conditions.

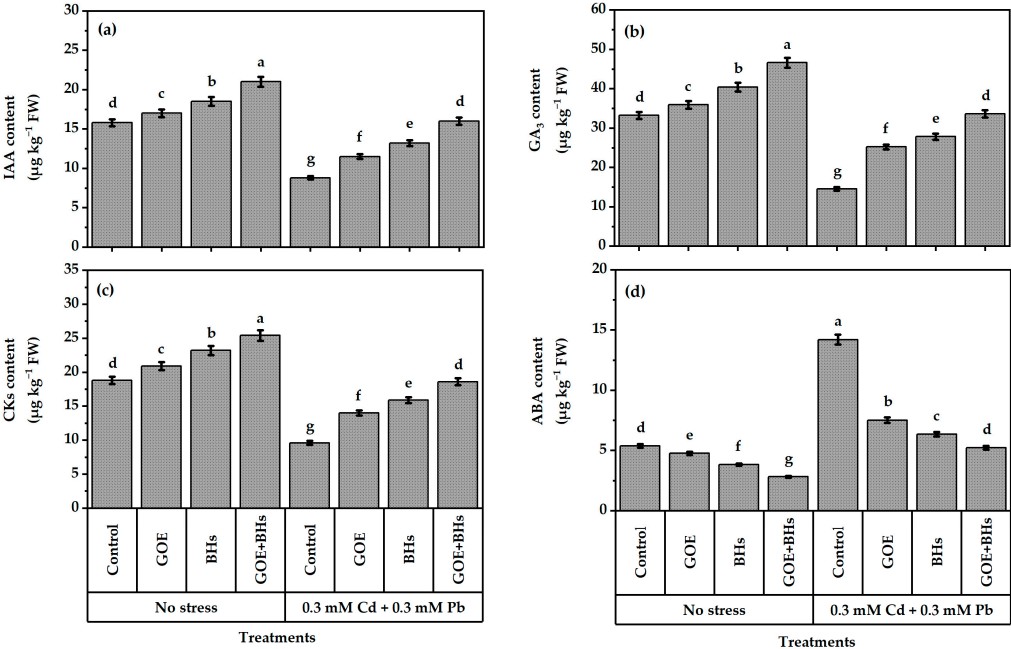

**Figure 8.** Responses of hormonal content ((**a**) indole-3-acetic acid; IAA, (**b**) gibberellic acid; GA₃, (**c**) cyto-kinins; CKs, and (**d**) abscisic acid; ABA) of *Cucurbita pepo* plants to foliar spraying with biostimulators (BSs) under normal or heavy metal (cadmium (Cd) + lead (Pb)) stress conditions. In each column, each two means followed by different letters indicate a significant difference according to the LSD test ($p \leq 0.05$). GOE; garlic + onion extract. BHs; bee-honey solution.

### 3.8. Growth and Yield Traits

As depicted in Figures 9 and 10, under normal conditions, foliar spraying of GOE, BHs or GOE + BHs significantly increased growth parameters (shoot length; ShL, number of leaves per plant; NoL, leaves area per plant; LsA, shoot fresh weight; ShFw, and shoot dry weight; ShDw) and yield traits (fruit number per plant; FNo, average fruit weight; AFw, and plant fruit yield; FY) of squash plants. The best treatment was GOE + BHs, which increased ShL, NoL, LsA, ShFw, ShDw, FNo, AFw, and FY by 24, 26, 57, 65, 80, 20, 7, and 29%, respectively, relative to the control. Combined (Cd + Pb) stress markedly decreased ShL, NoL, LsA, ShFw, ShDw, FNo, AFw, and FY by 27, 25, 59, 58, 55, 61, 65, and 86%, respectively, relative to the control. However, the application of different BSs (GOE, BHs, or GOE + BHs) restored the growth and yield traits of squash plants grown under Cd + Pb stress. The best treatment was GOE + BHs, which increased ShL, NoL, LsA, ShFw, ShDw, FNo, AFw, and FY by 38, 31, 149, 141, 133, 154, 185, and 626%, respectively, relative to the stressed control. The efficiency of the GOE + BHs treatment in restoring the growth and yield traits of squash plants under Cd + Pb stress was greater than that under normal conditions.

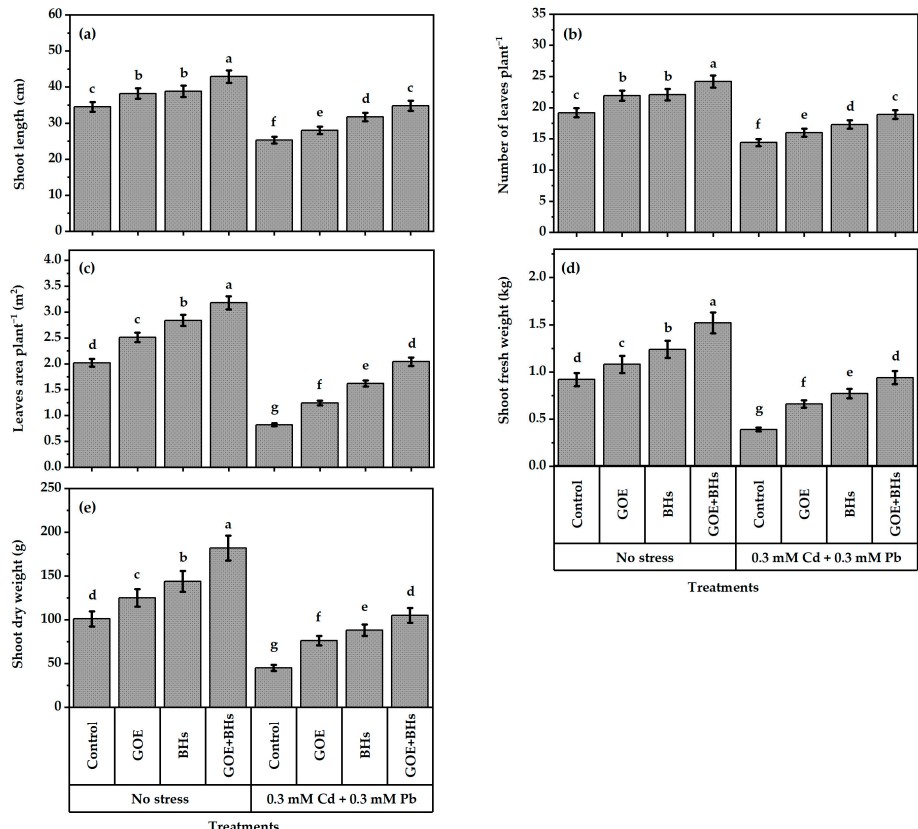

**Figure 9.** Responses of growth traits ((**a**) shoot length, (**b**) number of leaves per plant, (**c**) leaves area per plant, (**d**) shoot fresh weight, and (**e**) shoot dry weight) of *Cucurbita pepo* plants to foliar spraying with biostimulators (BSs) under normal or heavy metal (cadmium (Cd) + lead (Pb)) stress conditions. In each column, each two means followed by different letters indicate a significant difference according to the LSD test (*p* ≤ 0.05). GOE; garlic + onion extract. BHs; bee-honey solution.

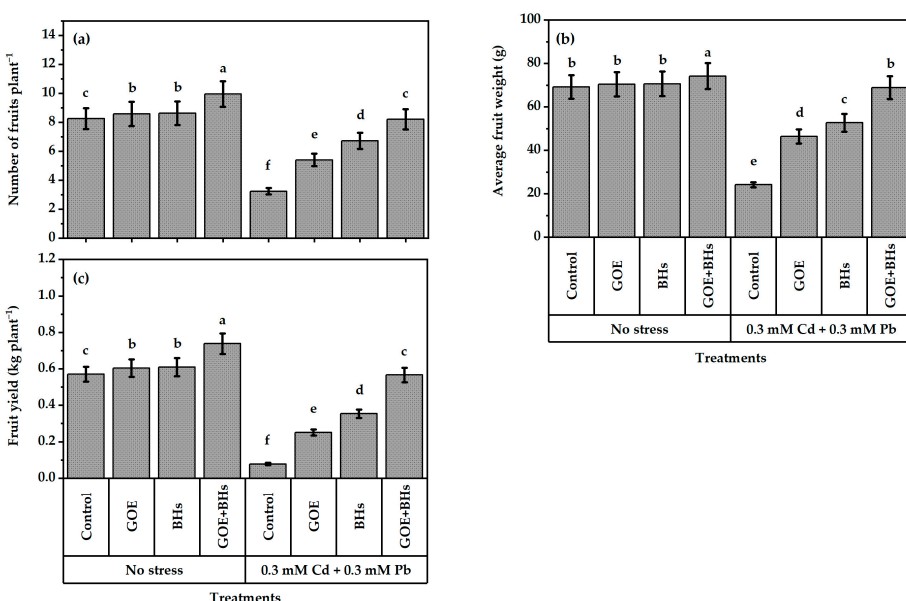

**Figure 10.** Responses of yield traits ((**a**) number of fruits per plant, (**b**) average fruit weight, and (**c**) plant fruit yield) of *Cucurbita pepo* plants to foliar spraying with biostimulators (BSs) under normal or heavy metal (cadmium (Cd) + lead (Pb)) stress conditions. In each column, each two means followed by different letters indicate a significant difference according to the LSD test (*p* ≤ 0.05). GOE; garlic + onion extract. BHs; bee-honey solution.

### 3.9. Fruit Quality Traits

As revealed in Figure 11, under normal conditions, foliar spraying of GOE, BHs, or GOE + BHs significantly improved the fruit quality traits of squash plants. The best treatment was GOE + BHs, which did not affect fruit acidity and increased vitamin C, total soluble solids (TSS), and soluble sugars by 22, 31, and 37%, respectively, while fruit Cd and Pb contents decreased by 33 and 27%, respectively, compared to the control. Combined (Cd + Pb) stress did not affect fruit acidity and decreased vitamin C, TSS, and soluble sugars by 27, 24, and 21%, respectively, while fruit Cd and Pb contents increased by 885 and 596%, respectively, compared to the control. However, the application of different GOE, BHs, or GOE + BHs restored the fruit quality traits of squash plants grown under Cd + Pb stress. The best treatment was GOE + BHs, which increased vitamin C, TSS, and soluble sugars by 41, 31, and 31%, respectively, while fruit Cd and Pb contents decreased by 92 and 87%, respectively, compared to the stressed control. The efficiency of the GOE + BHs treatment in restoring the fruit quality traits of squash plants under Cd + Pb stress was greater than that under normal conditions.

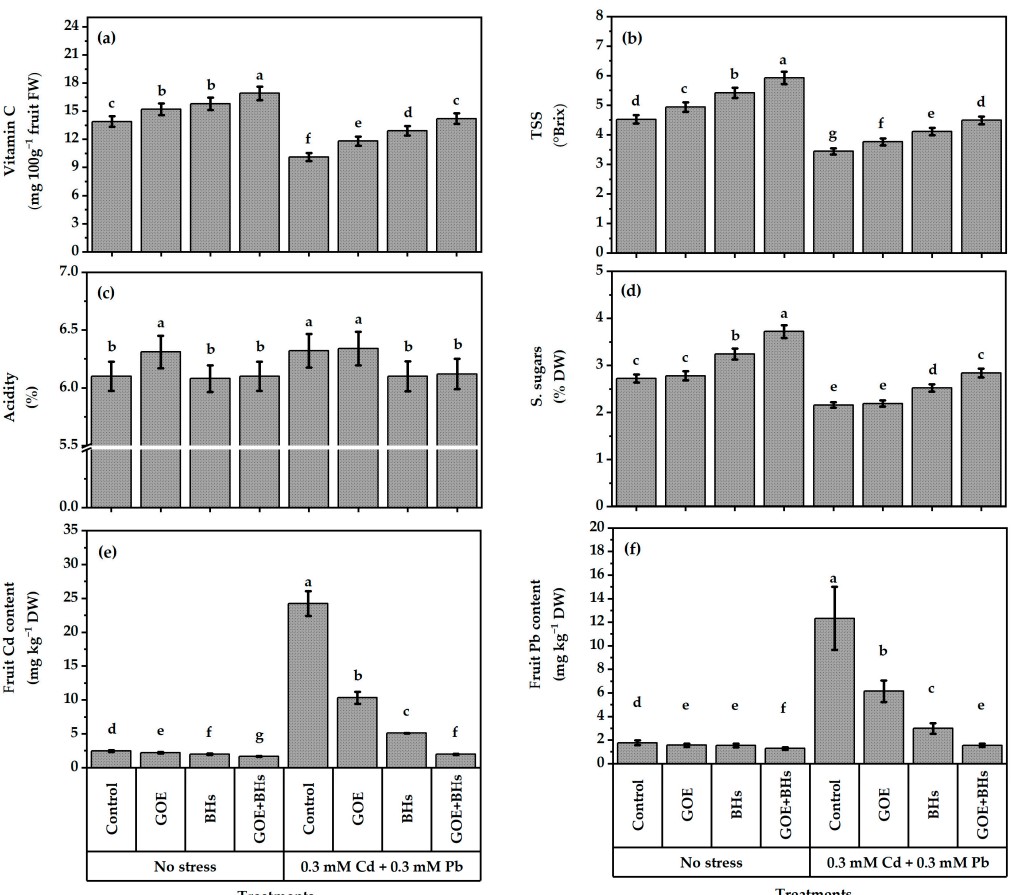

**Figure 11.** Responses of fruit quality traits ((**a**) vitamin C, (**b**) total soluble solids; TSS, (**c**) acidity, (**d**) soluble sugars, (**e**) fruit Cd content, and (**f**) fruit Pb content) of *Cucurbita pepo* plants foliar spraying with biostimulators (BSs) under normal or heavy metal (cadmium (Cd) + lead (Pb)) stress conditions. In each column, each two means followed by different letters indicate a significant difference according to the LSD test ($p \leq 0.05$). GOE; garlic + onion extract. BHs; bee-honey solution.

### 3.10. Relationships

Pearson's correlation analysis, in addition to hierarchical analysis, was conducted to test the relationships between all variables measured under leaf spraying with BSs (GOE, BHs, or GOE + BHs) and normal or Cd + Pb stress conditions (Figure 12). A positive correlation was explored significantly ($p \leq 0.05$) between FY, AFw, FNo, fruit vit C content,

and TSS with the contents of CKs, IAA, GA$_3$, soluble sugars, soluble protein, TChC, TCrC, PhA, Fv/Fm, PI, MSI, SPAD value, NoL, LsA, ShL, ShFw, and ShDw. Meanwhile, levels of EL, H$_2$O$_2$, O$_2^{\bullet-}$, MDA, contents of ABA, root Cd$^{2+}$, leaf Cd$^{2+}$, root Pb, leaf Pb, fruit Cd$^{2+}$, and fruit Pb had a negative correlation ($p \leq 0.05$) with the all above traits. A positive correlation was explored significantly ($p \leq 0.05$) between GB, GSH, and AsA contents, activities of CAT, SOD, APX, GR, $\alpha$ToC, free proline, and relative expression of GR, APX, SOD, and PrxQ with each other was also explored when applying Pearson's correlation analysis (Figure 12).

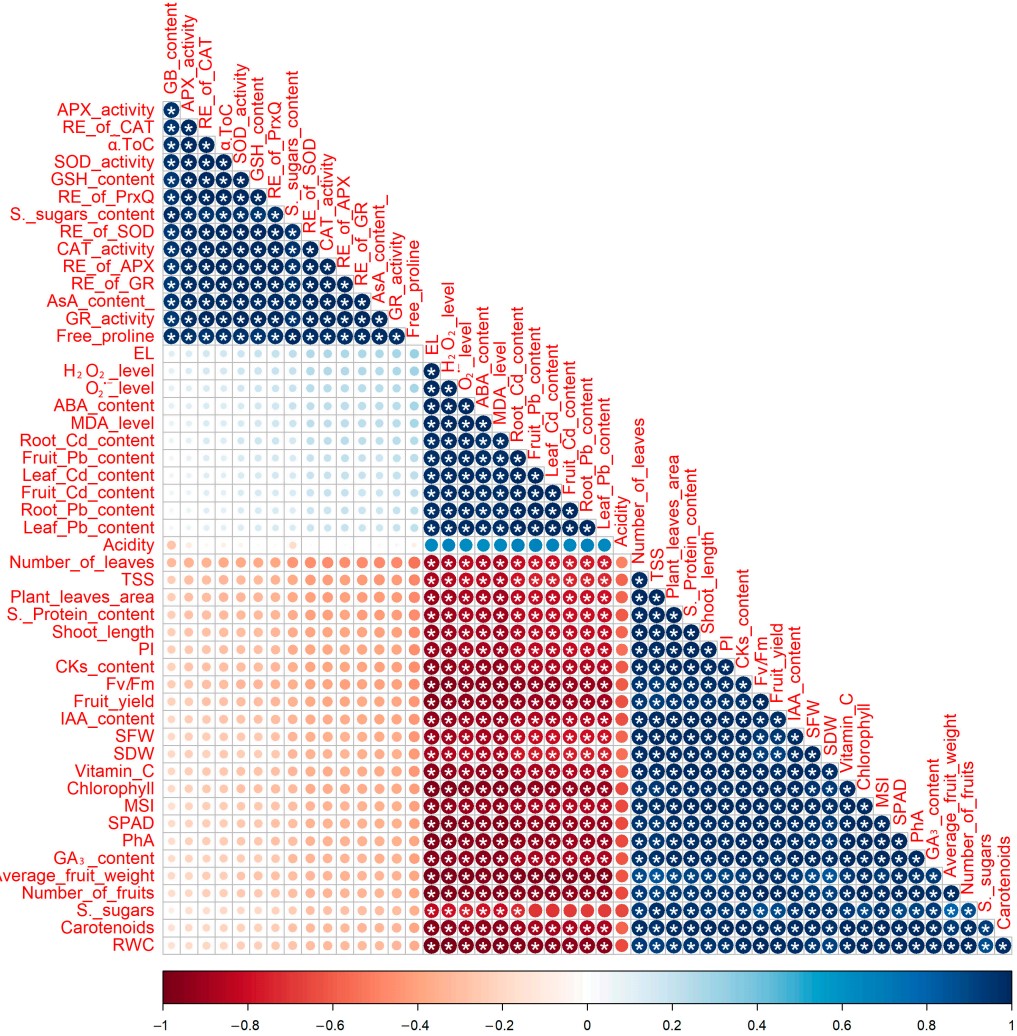

**Figure 12.** Pearson's correlation analysis among the different studied parameters. The colors represent variations in the Pearson's correlation value. * Indicates the significant at $p \leq 0.05$. SPAD value = Soil Plant Analysis Development, Fv/Fm = photosystem II quantum efficiency, RWC = relative water content, O$_2^{\bullet-}$ = Superoxide, H$_2$O$_2$ = Hydrogen peroxide, MDA = Malondialdehyde, EL = Electrolyte leakage, AsA = Ascorbate, and GSH = Glutathione, SOD = Superoxide dismutase, CAT = Catalase, APX = Ascorbate peroxidase, GR = Glutathione reductase, ABA = abscisic acid, GSH = glutathione, TOC = tocopherol, MSI= membrane stability index, TSS = total soluble solids. PI; performance index. PhA; Photochemical activity.

The hierarchical analysis split the study treatments into three main groups (Figure 13). The first group included no stress_BHs, no stress_GOE + BHs, no stress_control, and no stress_GOE, all revealed higher performance relative to the second group (Cd + Pb stress_BHs and Cd + Pb stress_GOE + BHs) and the third group (Cd + Pb stress_control and Cd + Pb stress_GOE). GOE + BHs achieved higher performance than GOE or BHs

under Cd + Pb stress and no stress conditions, thus the use of GOE + BHs mitigated the deleterious effects of Pb + Cd stress and improved growth and fruit quality of squash under no stress conditions.

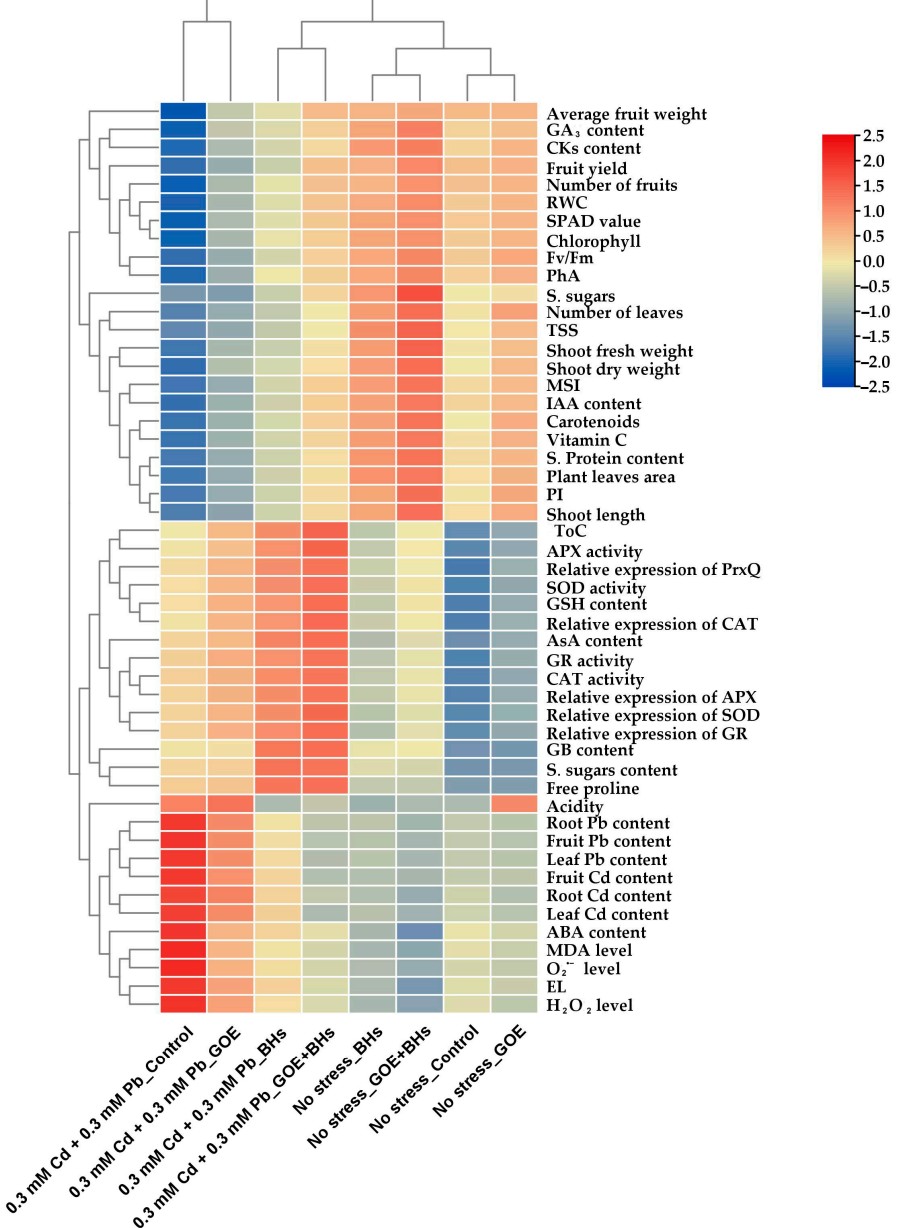

**Figure 13.** Heat map graph shows analysis of hierarchical clustering among the different studied parameters and treatments of leaf spray of *Cucurbita pepo* plants with biostimulators (BSs) under normal or heavy metal (cadmium (Cd) + lead (Pb)) stress conditions. The scale bar represents Z-score values of data of each parameter. SPAD value = Soil Plant Analysis Development; this numerical SPAD value specifies the relative content of chlorophyll within the leaf sample. Fv/Fm = photosystem II quantum efficiency, RWC = relative water content, $O_2^{\bullet-}$ = Superoxide, $H_2O_2$ = Hydrogen peroxide, MDA = Malondialdehyde, EL = Electrolyte leakage, AsA = Ascorbate, and GSH = Glutathione, SOD = Superoxide dismutase, CAT = Catalase, APX = Ascorbate peroxidase, GR = Glutathione reductase, ABA = abscisic acid, GSH = glutathione, TOC = tocopherol, MSI = membrane stability index, TSS = total soluble solids.

The principal component analysis (PCA) biplot showed significant variability in all parameters studied by foliar spraying of squash plants with bio-stimulants (BSs) under

normal or Cd+ Pb stress conditions. PCA-diminution 1 and 2 (Dim-1 and Dim-2, respectively) showed 68.5% and 26.2% of variability, respectively (Figure 14). The high variability between the no stress_GOE + BHs treatment and no stress_control treatment on the one hand and between the Cd + Pb_GOE + BHs and Cd + Pb_control treatments on the other hand indicated a role of GOE + BHs application in enhancing the growth parameters and physio-biochemical traits of squash under no stress and stress conditions. Foliar spraying with GOE + BHs enhanced soluble sugars, proteins, and proline contents, as well as CAT, GR, SOD, and APX activities, and their relative expressions. Moreover, the PCA-biplot indicated a positive impact of GOE + BHs treatments on the contents of $GA_3$, PhA, IAA, Fv/Fm, vit C, SPAD, and RWC in squash plants under no and Cd + Pb stress conditions with decreasing $H_2O_2$, MDA, $O_2^{\bullet-}$, Cd, and Pb levels in the squash roots, leaves, and fruits. Therefore, GOE + BHs treatment has a good role in improving the growth indices and overcoming the heavy metal stresses in squash plants.

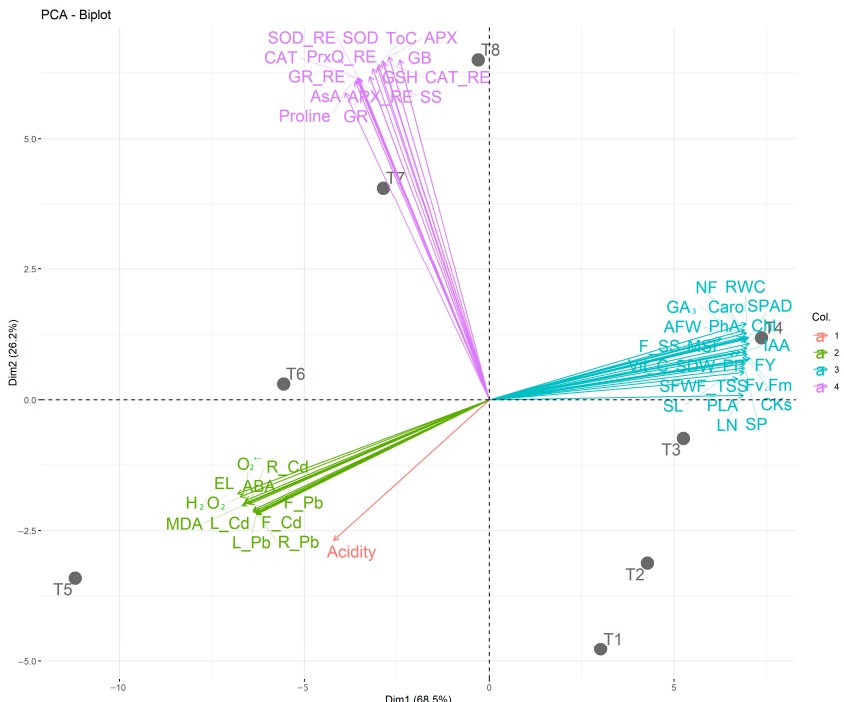

**Figure 14.** Bi-plot graph of studied parameters and treatments, showing the first two principal component analysis (PCA) dimensions (Dim1 and Dim2). LN = number of leaves, F_TSS = fruit total soluble solids, FY = Fruit_yield, AFW = Average_fruit_weight, NF = number of fruits per plant, SDW = shoot dry weight, SFW = shoot fresh weight, PLA = total plant leaves area, SL = shoot length, SP = S._protein_content, SS = S._sugars_content, Vit. C = vitamin C, RWC = relative water content, $O_2^{\bullet-}$ = Superoxide, $H_2O_2$ = Hydrogen peroxide, MDA = Malondialdehyde, EL = Electrolyte leakage, AsA = Ascorbate, GSH = Glutathione, ABA = abscisic acid, TOC = tocopherol, MSI = membrane stability index, TSS = total soluble solids. Caro = carotenoids, SOD = Superoxide dismutase, CAT = Catalase, APX = Ascorbate peroxidase, GR = Glutathione reductase, APX_RE = Ascorbate peroxidase relative expression, GR_RE = Glutathione reductase relative expression, Prxq_RE = Peroxiredoxin relative expression, IAA; indole-3-acetic acid, $GA_3$; gibberellic acid, CKs; Cytokinins, ABA; abscisic acid, and S. protein; soluble protein, L_Pb= Leaf lead, R_Pb= Root lead, L_Cd = leaf cadmium, R_Cd = Root cadmium, F_Cd = fruit cadmium, F_Pb = fruit lead, Fv/Fm; photosystem II quantum efficiency, PI; performance index, and PhA; Photochemical activity, GB; glycine betaine, S. sugars; soluble sugars, and αToC; α-Tocopherol. IAA; indole-3-acetic acid, $GA_3$; gibberellic acid, CKs; Cytokinins, T1 = No stress_Control, T2 = No stress_GOE, T3 = No stress_BHs, T4 = No stress_GOE + BHs, T5 = 0.3 mM Cd + 0.3 mM Pb_Control, T6 = 0.3 mM Cd + 0.3 mM Pb_GOE, T7 = 0.3 mM Cd + 0.3 mM Pb_BHs, T8 = 0.3 mM Cd + 0.3 mM Pb_GOE + BHs.

## 4. Discussion

As a result of various industrial and agricultural activities, the environment, including farmland, has been polluted by heavy metals (HMs), which can reach the plant through air, water, or soil, negatively affecting plant growth, yield, and crop quality. When HMs are present, root and shoot growth decrease, or photosynthesis stops, weakening respiration, thus negatively affecting vegetative and generative organs of plants [1,18]. The presence of HMs in more than the permissible value in plant tissues disrupts multiple metabolic events, such as transpiration, water and nutrient uptake, membrane stability, enzyme activity, stomatal movements, photosynthesis, protein synthesis, and disturbances of hormonal homeostasis [1,10,18]. As a result, plants develop and adopt their own self-defense systems as specific mechanisms and strategies to defend themselves against stress. Among them are ion hemostasis, osmotic adjustment, and stimulation of osmoregulation and antioxidant defense systems. However, these self-defense systems are insufficient in situations of prolonged stress.

Bio-stimulators (BSs) have been used effectively for stressed plants to promote osmoregulation and antioxidant defense systems, nutritional efficiency, productivity, and crop quality characteristics due to the improvement in the plant tolerance to stress [18–23,26–32]. As a novel strategy for HM-stressed squash plants, this report used a combined treatment incorporating garlic extract (GE) mixed with onion extract (OE) at a 1:1 (*v*/*v*) ratio (GOE), all with a diluted bee honey solution (BH), as natural inexpensive, and eco-friendly materials. These materials (GE, OE, and BHs) when applied individually noticeably improved plant growth, productivity, and crop quality due to improvements in osmoregulation and antioxidant defense systems in *Atriplex nummularia*, Anna apple trees, faba bean, onion, and common bean plants under abiotic stresses [20,26–32]. Therefore, the use of the combined (GOE + BHs) treatment was expected to be more efficient for Cd + Pb-stressed squash plants in this study.

The HM (Cd + Pb) stress decreased the SPAD value, photosynthetic efficiency (Fv/Fm, PI, and phytochemical activity; PhA) and synthesis and the accumulation of photosynthetic pigments (Figure 1) [68]. This result may be due to reduced leaf stomatal conductance, transpiration, relative water content (RWC), and increased osmotic stress in plants, which causes physiological damage [69]. Under Cd stress, the photosynthetic system and pigments are impaired by reducing chlorophyll production, inhibiting enzymes involved in $CO_2$ fixing, and negatively affecting cellular divisions and chloroplasts [69–71]. In addition, Pb negatively affects the photosynthetic pathways of plants because it destroys the chloroplast infrastructure and inhibits the production of key pigments, including chlorophyll, carotenoids, and plasto-quinones. It also hinders the Calvin cycle and the electron transport chain while reducing $CO_2$ by closing tomatal pores [13]. However, foliar nutrition with BSs (GOE, BHs, or GOE + BHs) improved the photosynthetic efficiency and pigment synthesis under normal conditions and restored the photosynthetic machinery and upregulated the photosynthetic pigment products under HM stress (Figure 1). This positive result may be due to the protective effects of GOE, BHs, or GOE + BHs, due to their containing of soluble sugars and other osmo-protectors (OPs), different antioxidants, and mineral nutrients (Tables S3 and S4), on the photosynthetic systems under normal or Cd + Pb stress conditions, improving plant defense against stress. The stimulatory effects of GOE extract on chlorophyll synthesis may be attributable to their active role in the synthesis of amino levulinic acid; the precursor of chlorophyll biosynthesis [72]. In addition, the sugars, alkaloids, flavonoids, terpenoids, ascorbate (AsA), and other antioxidants, along with mineral nutrients (Table S3) present in GOE can be effective in providing tolerance to stress and improving the photosynthetic efficiency [73]. The antioxidants, OPs, and nutrients in BHs are also plentiful, maintaining the intercellular hemostasis of the ions necessary for photosynthetic biosynthesis and improving the effectiveness of the photosynthetic apparatus in squash plants [30].

Cd + Pb toxicity negatively affected leaf RWC and membrane permeability (MSI), resulting in a decrease in squash plant water content (Figure 2). The build-up of Cd ions in

tissues has been linked to a decrease in root water content and may affect the uptake of water from soil [74], resulting in water scarcity in different plant organs [69]. In addition, Pb toxicity is believed to be involved in the restriction of water uptake mediated by decreased root hydraulic conductivity, which can reduce cellular turgor and decrease RWC. Similarly, previous researchers have demonstrated that coercion of HMs can reduce RWC of several plants, including pepper (cv. Semerkand), rocket, and maize (cvs. Syngenta-8711 and Syngenta-33H-25) [68,75,76]. However, BS treatments supported the maximal OP contents (Figure 5) in squash plants to conduct optimal RWC preservation upon cell requirement, which increased water conservation in plants under Cd + Pb stress. Figure 2 shows a significant relationship between the rise in RWC of squash plants and an improvement in plant hydration status brought on by BHs treatments. The applications of GOE, BHs, or GOE + BHs under Cd + Pb stress boosted RWC and preserved healthy metabolic functions in squash. The fact that GOE + BHs helped maintain the integrity of cell membranes (Figure 2) and allowed plants to develop and perform well under normal and Cd + Pb stress conditions may be the cause of these beneficial changes in the membranes of leaf tissue and RWC.

As depicted in Figure 3, higher Pb and Cd contents were observed as a result of the application of Cd + Pb treatment in plant roots relative to the aerial parts, which has been previously confirmed [77,78]. However, three processes may be functioning and leading to enhanced Pb and Cd sequestration in plant roots; (1) precipitation of insoluble Pb in the intercellular gaps, (2) sequestration of the metal in the vacuoles of rhizodermal and cortical cells, and (3) fixation of the metal by negatively charged pectin in cell walls [75,79]. The application of BSs, especially GOE + BHs, markedly limited the accumulations of Cd and Pb contents of roots and leaves of squash plants under normal and Cd + Pb stress conditions (Figure 3). This positive result can be obtained due to the presence of several nutrients (some of which may antagonize Cd and Pb), OPs (soluble carbohydrates and proline) and various antioxidants in GOE and BHs (Tables S3 and S4). As reported [80], BSs increase plant vigor and raise plant tolerance and phytoremediation activity against HMs stress. The results of Alharby et al. [10], Abou-Sreea et al. [22], Semida et al. [31], Bartucca et al. [80], and Rady et al. [81] confirmed those in this study.

In this study, the highly increased root and leaf contents of Cd and Pb due to Cd + Pb stress contributed to the increase in oxidative stress markers ($H_2O_2$ and $O_2^{\bullet-}$) and their damage (electrolyte leakage; EL and lipid peroxidation measured as malondialdehyde; MDA) (Figure 4), negatively affecting several aspects related to plant metabolism [80–82]. The ROS induces MDA and EL. The MDA is a significant indicator of oxidative stress resulting from membrane disruption [82], further impairing the proper function of cells and negatively affecting plant development and production [83]. A chain reaction of lipid peroxidation starts when ROS molecules target polyunsaturated fatty acids in cell membrane lipids, changing the cell membrane structure and function. EL is widely considered a fundamental measure of the permeability status of cell membranes. In the present study, increased EL caused by elevated levels of $H_2O_2$ and $O_2^{\bullet-}$ (caused by Cd + Pb stress) severely damaged squash plants (Figure 4). However, foliar-supplementation of GOE, BHs, or GOE + BHs, especially GOE + BHs, significantly decreased the accumulation of these ROS and increased squash leaf integrity (minimized MDA and EL levels), which may improve plant different metabolic reactions under Cd + Pb stress. Besides, BSs application helps to increase the sequestration of HMs in different plant parts [80] to relieve their negative impacts. GOE + BHs delivered enhanced protective inducers, such as mineral nutrients, OPs, and various antioxidants, which shielded plasma membranes from lipid peroxidation by lowering $H_2O_2$ and $O_2^{\bullet-}$ levels. These distinct properties are essential powerful mechanisms to prevent lipid peroxidation in cell membranes, raise plant water content, and minimize oxidative stress-caused damage under stress conditions. These results may be attributable to decreased EL and photo-oxidation, enhanced MSI, and induced membrane integrity against oxidative damage, hence resulting in enhanced squash plant growth and yield under Cd + Pb stress. GOE is rich in flavonoids and organosulfur

compounds (anthocyanin and quercetin) [27,28], which contribute to the protection of cell membranes. Detoxification systems are also modulated by flavonoids and organosulfur compounds, which also have antioxidant and metal-chelating characteristics. In addition, GOE is rich in calcium, which prevents cell membrane damage and permeability and stabilizes cell membrane structure under adverse environmental conditions [72].

Relative to the control plants, the quantities of OPs, including glycine betaine (GB), proline, and soluble sugars, markedly increased under Cd + Pb stress (Figure 5). Numerous plant species commonly respond physiologically to HMs poisoning by increasing GB, soluble sugars, and proline [11,79,84]. These metabolites can substantially boost a plant's general tolerance to stress. The two most important metabolic functions of soluble carbohydrates are osmotic adjustment and the protection of cellular components. Similar functions have been postulated for proline in the cell because it may actively participate in Cd and Pb inactivation and complex formation. Proline is a non-essential amino acid that is synthesized in cell chloroplasts and cytoplasm. For plant adaptation and tolerance to HMs stress, proline accumulates in plants as a plant policy for mitigating HMs stress through modification of cell osmosis, stabilization of membrane structures, and minimization of oxidative stress damage [11]. The non-enzymatic antioxidant defense system includes AsA and GSH activation to control cellular redox and stimulates the cell production of $\alpha$ToC, carotenoids, and phenolic compounds (PhCs). The PhCs and redox regulators integrate with components of the plant's defense mechanism. GSH also serves a crucial function in protecting cell membranes from oxidative stress caused by ROS. Enzymatic antioxidants neutralize $H_2O_2$ by converting it into nontoxic substances ($H_2O$ and $O_2$) [84]. Because cellular GSH is susceptible to oxidative stress, it takes action as the first defense line in cases of Cd and Pb poisoning. By keeping tocopherol and zeaxanthin in reduced forms, GSH indirectly contributes to the protection of membranes. Protein denaturation brought on by the oxidation of protein thiol groups during HMs stress is prevented by GSH. Although Cd + Pb stress increased OPs and non-enzymatic antioxidant contents, the application of GOE, BHs, or GOE + BHs, especially GOE + BHs, further increased their contents to support the osmoregulation and antioxidant defense systems against Cd + Pb stress. Application of GOE + BHs contributed to the increase in OPs and non-enzymatic antioxidants under normal conditions by 13–40% and under Cd + Pb stress by 21–46%. This positive result may be due to the richness of these BSs in the mineral nutrients, OPs, and non-enzymatic antioxidants, which easily penetrate stomata, after foliar spraying, into leaf tissues [20,26–31]. The improving effect of these BSs was more pronounced under stress than under no stress conditions. This could be due to the plant's need to increase the strength of its defenses against stress.

As depicted in Figure 6, Cd + Pb stress notably enhanced the antioxidant enzyme activities (SOD; superoxide dismutase, CAT; catalase, APX; ascorbate peroxidase, and GR; glutathione reductase), enhancing the tolerance of squash plants. This result may be attributable to the increased toxicity caused by the Cd + Pb and the increased activities of antioxidant enzymes function to protect plants from Cd and Pb toxicity. Therefore, the plant's defense mechanism becomes more responsive when multiple metals are applied simultaneously [79]. When oxidative stress occurs, SOD, GR, CAT, and APX are examples of antioxidant enzymes that activate, along with increased contents of OPs and non-enzymatic antioxidants. Similar to OPs and non-enzymatic antioxidants, the development and adoption of enzyme activities increased further in GOE + BHs-treated plants (Figures 5 and 6). This result supports the osmoregulation, and enzymatic and non-enzymatic antioxidant defense systems to effectively suppress ROS under Cd + Pb stress. GOE and BHs are rich in mineral nutrients, OPs, and different antioxidants that are likely to be absorbed into leaf tissues. In addition, as reported before [31], glucose oxidase reacts with glucose in BHs to generate gluconic acid and $H_2O_2$, both of which have known antibacterial properties. Honey's $H_2O_2$ is oxidized by CAT, releasing nascent oxygen. The antioxidant and scavenging properties of BHs have been confirmed [30]. Flavonoids are also important functional components.

Our study examined the enzyme genes expressions (*SOD*, *GR*, *APX*, *CAT*, and *PrxQ*) in squash plants to identify BSs-mediated tolerance (Figure 7). All enzyme gene transcripts were substantially expressed in plants growing under Cd + Pb stress and were expressed more in stressed squash plants inoculated with BSs, indicating that enzyme activation occurred (Figure 6). Previous studies [10,22,33] showed that all enzyme activities examined were significantly increased with the increase in enzyme gene expressions, strengthening defense systems in stressed plants. This strategy takes place in plants under a variety of stresses, including Cd + Pb.

The action mechanisms of various hormones for distinct functions might vary greatly. Consequently, one hormone may govern several processes related to the cell and its development, whereas numerous hormones may be engaged in regulating one activity. IAA is a multifunctional hormone responsible for plant growth development under normal and stressful conditions. ABA contributes to several plant developmental and physiological processes, which include stomatal closure, embryo maturation, seed dormancy stimulation, and stress responses [85]. A quick increase in endogenous ABA levels has been seen in response to environmental stressors, which activate certain signaling pathways and alter gene expression levels in the plant. As shown in Figure 8, exposure to Cd + Pb markedly raised endogenous ABA levels, while decreasing IAA, $GA_3$, and CKs. ABA may operate as a trade-off between plant responses to HMs-induced stress, triggering a balance between survival and growth. By applying BSs, especially GOE + BHs, cyto-kinins (CKs) were significantly increased along with an increase in indole-3-acetic acid (IAA) and gibberellic acid ($GA_3$), under normal and Cd + Pb stress conditions, to enable plants to withstand stress (Figure 8). CKs antagonize ABA, regulate plant development, and participate intensely in interactions with other hormones under HMs stress [86]. $GA_3$ promotes seed germination, plant growth, and fruit development. It also promotes the adaptability and tolerance of plants to several stressors and reduces the toxicity of HMs. The increase in phytohormone contents, due to the foliar application of GOE + BHs, mitigate and repair damage caused by ROS under stress along with antioxidants [22,87]. At modest concentrations, BHs in GOE + BHs treatment, glucose oxidase catalyzes the oxidation of glucose to release $H_2O_2$ in low concentration to protect squash against Cd + Pb stress and increased phytohormone contents to contribute to the increase of squash plant tolerance and performance under HM stress [22,31].

In the present study, Cd + Pb stress negatively affected plant growth, development, yield, and fruit quality (Figures 9–11). Cd stress causes a marked reduction in leaf area and plant dry weight [88]. Growth of alfalfa [89], wheat (cv. (Lasani-2008) [71]), and spinach [90] is inhibited by high Cd concentrations. Pb poisoning negatively affects water hyacinths and greatly stunts plant development. Pb accumulates mostly in root tissues, followed by petiole and leaf tissues [13]. Several studies have also indicated that the suppression of plant development under Cd and/or Pb may be connected with the water transport system in plant tissues [91]. In addition, the loss in biomass may be related to the mineral's interference with micronutrient uptake and the oxidative stress it causes, which inhibits root cell proliferation and increases lignification [92]. It is common for HMs to directly or indirectly inhibit a number of physiological processes, including transpiration, respiration, photosynthesis, N metabolism, cell elongation, and nutrient uptake. As a result, growth is slowed, leaves turn chlorosis, and biomass is reduced [67,93]. According to our data, Cd + Pb greatly reduced the fresh and dried shoot weights of squash; however, fortifying plants with BSs reversed this effect and significantly improved plant growth metrics. It was also observed that squash, which effectively received BS components that penetrated the leaves after spraying the feeding solution, had a high tolerance towards Cd + Pb stress. Our findings corroborate the results of Alharby et al. [10] and Rady et al. [30]. BSs (GOE + BHs)-enhanced root development, leaf RWC, and photosynthesis are the direct causes of the enhancement of the growth and yield of squash plants under stressful conditions [10,22,33]. Previous research has demonstrated that BSs improve plant yield and quality, even when plants are stressed [30,82,94]. This outcome, in this study, may be

explained by the GOE's high amount of sulfur-containing amino acids like methionine and cysteine, which are crucial for plant biological processes [72]. In addition, the positive impacts of GOE on growth, physiological characteristics, anatomical structure, and yield components of stressed pea plants are also noted [26]. Using faba bean crop, Rady et al. [30] noted that BHs ameliorated the adverse impacts of stress conditions on the plant growth and yield. Our results showed that the bioactive GOE + BHs elements, including OPs (sugars, proline, and GB), antioxidants, and organic nutrients promoted plant adaptability to adverse stress conditions by enhancing cell division, elongation, and metabolism, as well as dry matter accumulation, thus enabling plants to withstand the negative effects of stress and significantly promote plant performance and yield. Thus, GOE + BHs may release bio-stimulatory compounds, which positively affect squash plant growth under Cd + Pb stress conditions. Finally, this study signalized that the pronounced improvements induced by GOE + BHs for photosynthetic efficiency, leaf integrity, OPs, enzymatic and non-enzymatic activities, enzyme gene expressions, and phytohormones, which were coincident with suppression of ROS, MDA, EL, and HMs (Cd and Pb), all contributed to increasing growth, yield, fruit quality traits of squash plants under normal or Cd + Pb stress conditions.

## 5. Conclusions

Foliar spraying of a novel combined treatment incorporating garlic extract mixed with onion extract at a 1:1 (*v*/*v*) ratio, all with a diluted bee honey solution (GOE + BHs) attenuated toxicity from the cadmium + lead stress application by improving growth, production, yield quality, photosynthetic efficiency, mineral nutrient contents, leaf tissue integrity, contents of osmotically-active substances, enzymatic and non-enzymatic antioxidant activities, transcript levels of enzyme genes, and phytohormone contents, thus improving antioxidant defense systems and triggering more scavenging reactive oxygen species to help squash plants survive under cadmium + lead stress. Rich in antioxidants, osmotically-active substances, nutrients, and vitamins, GOE + BHs treatment has been explored as an efficient ecofriendly strategy to attenuate the impacts of cadmium + lead stress. Using GOE + BHs as plant-based anti-stress support will help in agriculture and environmental sustainability. Although the individuals of GOE + BHs have previously been applied to some crop plants under some abiotic stresses, there is still a need for more research on different crop plants in order to uncover the precise mechanisms of GOE + BHs treatment with the objective of raising its potential to encourage farmers to use these natural bio-stimulators as a strategy to improve plant productivity and yield quality by stimulating the activity of the antioxidant defense systems and enhancing HMs stress tolerance in plants. However, some future challenges may arise with the use of GOE + BHs as an application strategy for plants. Although no pathological problems have been explored using GOE + BH treatment for stressed squash plants, extensive studies need to be conducted in this regard on different crop plants.

**Supplementary Materials:** The following supporting information can be downloaded at: https://www.mdpi.com/article/10.3390/agronomy13071916/s1. Table S1. Average weather data of the Fayoum District during the study season 2022, Table S2. Chemical composition of irrigation water, Table S3. Proximate composition of garlic + onion extract (on dry matter basis), Table S4. Major ingredients of raw clover honey (based on fresh weight) and Table S5. Primers sequences for semi-quantitative and quantitative RT-PCR of the stress-related genes in *C. pepo* plant.

**Author Contributions:** Conceptualization, M.M.R., M.M.M.S. and A.E.M.M.; methodology, M.M.R., E.F.A., S.K., A.K., K.A.S., A.E.M.M., A.F.A. and H.A.F.; software, E.F.A., M.M.M.S., S.K., A.K. and A.S.O.; validation, A.S.O., K.A.S., A.E.M.M., A.F.A. and H.A.F.; formal analysis, M.M.R., E.F.A. and M.M.M.S.; investigation, M.M.R., E.F.A., A.F.A. and H.A.F.; resources, A.K., A.S.O. and K.A.S.; data curation, M.M.R., E.F.A. and A.E.M.M.; writing—original draft preparation, M.M.R., E.F.A., M.M.M.S., S.K., A.K., A.S.O., K.A.S., A.E.M.M., A.F.A. and H.A.F.; writing—review and editing, M.M.R., E.F.A., S.K., A.K., A.S.O., K.A.S., A.E.M.M. and A.F.A.; visualization, A.E.M.M. and H.A.F.; supervision, M.M.R.; project administration, M.M.R. and E.F.A.; funding acquisition, A.F.A. and E.F.A.; All authors have read and agreed to the published version of the manuscript.

**Funding:** This research was funded by the Deanship of Scientific Research, Taif University.

**Institutional Review Board Statement:** Not applicable.

**Data Availability Statement:** The data presented in this study are available upon request from the corresponding author.

**Acknowledgments:** The researchers would like to acknowledge Deanship of Scientific Research, Taif University for funding this work.

**Conflicts of Interest:** The authors declare no conflict of interest.

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
