# Peer review of "Exploring the Role of Novel Biostimulators in Suppressing Oxidative Stress and Reinforcing the Antioxidant Defense Systems in Cucurbita pepo Plants Exposed to Cadmium and Lead Toxicity"

_agronomy, doi:10.3390/agronomy13071916_

Round 1

Reviewer 1 Report

This manuscript investigated the potential of new biostimulants based on plant extract and honey to mitigate oxidative stress causing by Cd and Pb and its possible mechanisms. The content of the manuscript fits well with the aim and scope of the journal and should attract a wide readership. Overall quality of the manuscript is good. The manuscript is well written and clearly explained. Therefore, I recommend a minor revision of this manuscript in its current format. Some concerns are listed below.

Introduction

1. Please add information regarding microbial biostimulants and their limitations to support why use plant extract as biostimulant is preferable.

2. Why investigate Cd and Pb? Are these 2 HMs affected C. pepo producing area?

3. What are the sources of these 2 HMS?

4. Why investigate effect of Cd+Pb and not Cd or Pb alone?

5. Line 79, 93 please delete etc

Materials and methods

Line 142 what is MM?  

Line 145 what is FF?

Line 177 delete “with”

Line 246 please add full name of IAA, GA and CK?

Line 253 what is “MSTFA”?

Line 256 what is LRI? 

Results and discussions

Line 330 “As shown, the effect of BSs was most pronounced under stress relative to no stress conditions”

Figure 4 and 5 were not referred to in the text.

It would be nice to examine the cost of the two biostimulant?

Line 359-361 Please explain why under stress-free conditions, GOE, BHs, or GOE+BHs treatment enhanced GB, soluble sugars, proline, AsA, GSH, and αToC levels by 13–40% relative to the corresponding controls

Quality of English is good with minor improvement needed

Author Response

Agronomy - MDPI

Manuscript ID: agronomy-2505322

Manuscript Title:  "Exploring the role of novel biostimulators in suppressing oxidative stress and reinforcing the antioxidant defense systems in Cucurbita pepo plants exposed to cadmium and lead toxicity"

==================================================

Dear Berry Song

Assistant Editor, Agronomy

         Thank you for your efforts and we would also like to thank the reviewers a lot for their valuable comments that helped improve our manuscript. We have corrected the manuscript based on the reviewers' comments, corrections made in the text in red, and outlined step by step as follows:

Response to the comments of Reviewer#1:

Introduction

  1. Please add information regarding microbial biostimulants and their limitations to support why use plant extract as biostimulant is preferable.

Re: Added (please see lines 110-120).

  1. Why investigate Cd and Pb? Are these 2 HMs affected C. pepo producing area?

Re: Cd and Pb are among the most polluted metals in Egyptian soil, including the studied area, resulting from industrial and agricultural activities and car exhausts along agricultural highways (please see lines 64-66).

  1. What are the sources of these 2 HMS?

Re: Cd and Pb are among the most polluted metals in Egyptian soil, including the studied area, resulting from industrial and agricultural activities and car exhausts along agricultural highways (please see lines 64-66).

  1. Why investigate effect of Cd+Pb and not Cd or Pb alone?

Re: Cd and Pb are among the most polluted metals in Egyptian soil, including the studied area, resulting from industrial and agricultural activities and car exhausts along agricultural highways (please see lines 64-66).

  1. Line 79, 93 please delete etc

Re: etc has been deleted (please see lines 85 and 104).

Materials and methods

Line 142 what is MM? 

Re: Revised to modified medium (please see lines 181, 185, 187, 188, and 192).

Line 145 what is FF?

Re: Revised to fertilizer formulation (please see line 191-192).

Line 177 delete “with”

Re: "with" has been deleted (please see line 225).

Line 246 please add full name of IAA, GA and CK?

Re: Added (please see lines 294-295).

Line 253 what is “MSTFA”?

Re: Revised to N-Methyl-N-(trimethylsilyl)trifluoroacetamide (please see line 302).

Line 256 what is LRI?

Re: Revised to linear retention indices (please see line 305).

Results and discussions

Line 330 “As shown, the effect of BSs was most pronounced under stress relative to no stress conditions”

Re: Revised (please see line 382-383).

Figure 4 and 5 were not referred to in the text.

Re: The figures have been referred (please see lines 391 and 412).

It would be nice to examine the cost of the two biostimulant?

Re: This study was done using pots, and it would certainly be good to examine the cost of the biostimulants used when doing the experiments at the field level then the cost would be accurate. But it sure would be fine to obtain a product free from heavy metal contamination.

Line 359-361 Please explain why under stress-free conditions, GOE, BHs, or GOE+BHs treatment enhanced GB, soluble sugars, proline, AsA, GSH, and αToC levels by 13–40% relative to the corresponding controls.

Re: Explained (please see lines 750-756).

Many thanks to Reviewer#1 for his valuable comments

Esmat F. Ali (Corresponding author)

Mostafa M. Rady (Corresponding author)

Reviewer 2 Report

Very extensive and comprehensive research, rarely seen in the evaluation of scientific publications. Only in the text of the publication the places to provide details were indicated. In the summary, give the values ​​of changes in the most important parameters in percent. Throughout the publication, where you can give specific species of plants that other authors have studied. In this way, this publication will be perfect. 

Good level of English

Author Response

Agronomy - MDPI

Manuscript ID: agronomy-2505322

Manuscript Title:  "Exploring the role of novel biostimulators in suppressing oxidative stress and reinforcing the antioxidant defense systems in Cucurbita pepo plants exposed to cadmium and lead toxicity"

==================================================

Dear Berry Song

Assistant Editor, Agronomy

         Thank you for your efforts and we would also like to thank the reviewers a lot for their valuable comments that helped improve our manuscript. We have corrected the manuscript based on the reviewers' comments, corrections made in the text in brown, and outlined step by step as follows:

Response to the comments of Reviewer#2:

In the summary, give the values ​​of changes in the most important parameters in percent.

Re: Amended (please see abstract section).

Throughout the publication, where you can give specific species of plants that other authors have studied.

Re: Amended (please see lines 88-91, 94-95, 101-102, 105, 672-673, and 809).

All other requests (in the PDF) have been positively modified in the text in brown.

Many thanks to Reviewer#2 for his valuable comments

Esmat F. Ali (Corresponding author)

Mostafa M. Rady (Corresponding author)

Reviewer 3 Report

Review Report

Title: Exploring the role of novel biostimulators in suppressing oxidative stress and reinforcing the antioxidant defense systems in Cucurbita pepo plants exposed to cadmium and lead toxicity

In this paper, the authors investigated the role of novel biostimulators in improving the heavy metal tolerance in Cucurbita pepo by modifying the antioxidants defense mechanism. The work provided novel information to understand the mechanism against cadmium and lead toxicity with novel biostimulators treatment. In general, this manuscript shows well design and data and merits publication in Agronomy. However, there are still some minor issues that need to be revised prior to publication. Some descriptions need to be concise and clarify.

Suggestions/Comments to authors:
1. The language of the article is generally fluent, but some grammar usage should be checked

again.

2. In the abstract, please mention the levels of concentration of biostimulants.

3. Add the solid conclusion at the end of the abstract section
4.The introduction of the article needs to further sort out the latest literature related to these natural biostimulants and its role in heavy metals tolerance to plants.

5. Hypothesis of the study is missing; Authors should provide a clear research question

6. In the objectives (line 120-124); numbering must be removed  

7. How much volume of each biostimulants solution was used for each pot?

8. Please expand the statistical section, write in detail SE (Standard error), detail of turkeys test and ANOVA is two way not one way (two factors are studies 1. Stress levels and 2. Biostimulants application) and level of significance

9. I think the level of significance in control condition must be 1% instead of 5%

10. Results and discussion section of the article is weak, authors mainly focused on their results but they did not discuss them according to international standards. Moreover the writing style of results and discussion section is also ambiguous, with long and weak sentences and in a repetitive way. I am not convinced with the way of discussion of the authors, in its current form it cannot be accepted in agronomy. I will recommend a thorough revision of this section.

11. Quality of figures should be improved. They must be uniform in format, letter font and size should be the same as the remaining manuscript body. 

12. The conclusions should answer the hypothesis of your study and should focus on the implication of your findings. Please, avoid using abbreviations and acronyms in this section

13. The reference of the article needs to be checked, revised and formatted.
14. The reference of the article needs to be checked, revised and formatted.

Language, wording and paraphrasing should be carefully reviewed and improved. A native English-speaking scientist or professional English editing service must edit your manuscript.

Author Response

Agronomy - MDPI

Manuscript ID: agronomy-2505322

Manuscript Title:  "Exploring the role of novel biostimulators in suppressing oxidative stress and reinforcing the antioxidant defense systems in Cucurbita pepo plants exposed to cadmium and lead toxicity"

==================================================

Dear Berry Song

Assistant Editor, Agronomy

         Thank you for your efforts and we would also like to thank the reviewers a lot for their valuable comments that helped improve our manuscript. We have corrected the manuscript based on the reviewers' comments, corrections made in the text in blue, and outlined step by step as follows:

Response to the comments of Reviewer#3:

Suggestions/Comments to authors:

  1. The language of the article is generally fluent, but some grammar usage should be checked again.

Re: The manuscript has been grammatically checked by a native English-speaking scientist.

  1. In the abstract, please mention the levels of concentration of biostimulants.

Re: Amended (please see lines 25-26).

  1. Add the solid conclusion at the end of the abstract section

Re: Added (please see lines 44-46).

4.The introduction of the article needs to further sort out the latest literature related to these natural biostimulants and its role in heavy metals tolerance to plants.

Re: Amended (please see lines 125-134, 140-150).

  1. Hypothesis of the study is missing; Authors should provide a clear research question

Re: Amended (please see lines 162-168).

  1. In the objectives (line 120-124); numbering must be removed

Re: Removed (please see lines 160-163).

  1. How much volume of each biostimulants solution was used for each pot?

Re: Amended (please see lines 206-207).

  1. Please expand the statistical section, write in detail SE (Standard error), detail of turkey’s test and ANOVA is two way not one way (two factors are studies 1. Stress levels and 2. Biostimulants application) and level of significance

Re: Amended (please see lines 323-330).

  1. I think the level of significance in control condition must be 1% instead of 5%

Re: Revised (please see lines 327).

  1. Results and discussion section of the article is weak, authors mainly focused on their results but they did not discuss them according to international standards. Moreover the writing style of results and discussion section is also ambiguous, with long and weak sentences and in a repetitive way. I am not convinced with the way of discussion of the authors, in its current form it cannot be accepted in agronomy. I will recommend a thorough revision of this section.

Re: Results and discussion sections have been improved (please see the sections of Results and Discussion).

  1. Quality of figures should be improved. They must be uniform in format, letter font and size should be the same as the remaining manuscript body.

Re: Improved (please see Figures).

  1. The conclusions should answer the hypothesis of your study and should focus on the implication of your findings. Please, avoid using abbreviations and acronyms in this section

Re: Revised (please see Conclusions section).

  1. The reference of the article needs to be checked, revised and formatted.
  2. The reference of the article needs to be checked, revised and formatted.

Re: The reference list has been checked and formatted (please see the references list).

Comments on the Quality of English Language Language, wording and paraphrasing should be carefully reviewed and improved. A native English-speaking scientist or professional English editing service must edit your manuscript.

Re: The manuscript has been grammatically checked by a native English-speaking scientist.

Many thanks to Reviewer#3 for his valuable comments

Esmat F. Ali (Corresponding author)

Mostafa M. Rady (Corresponding author)
